# Maximum State Entropy Exploration using Predecessor and Successor Representations

**Arnav Kumar Jain**[*]
Mila–Quebec AI Institute
Université de Montréal

**Lucas Lehnert**[†]
Fundamental AI Research at Meta

**Irina Rish**
Mila–Quebec AI Institute
Université de Montréal

**Glen Berseth**
Mila–Quebec AI Institute
Université de Montréal

## Abstract

Animals have a developed ability to explore that aids them in important tasks such as locating food, exploring for shelter, and finding misplaced items. These exploration skills necessarily track where they have been so that they can plan for finding items with relative efficiency. Contemporary exploration algorithms often learn a less efficient exploration strategy because they either condition only on the current state or simply rely on making random open-loop exploratory moves. In this work, we propose $\eta\psi$-Learning, a method to learn efficient exploratory policies by conditioning on past episodic experience to make the next exploratory move. Specifically, $\eta\psi$-Learning learns an exploration policy that maximizes the entropy of the state visitation distribution of a single trajectory. Furthermore, we demonstrate how variants of the predecessor representation and successor representations can be combined to predict the state visitation entropy. Our experiments demonstrate the efficacy of $\eta\psi$-Learning to strategically explore the environment and maximize the state coverage with limited samples.

## 1 Introduction

Animals and humans are very efficient at exploration compared to their data-hungry algorithms counterparts [36, 28, 50, 51, 12, 14, 65, 52]. For instance, when misplacing an item, a person will methodically search through the environment to locate the lost item. To efficiently explore, an intelligent agent must consider past interactions to decide what to explore next and avoid re-visiting previously encountered locations to find rewarding states as fast as possible. Consequently, the agent needs to reason over potentially long interaction sequences, a space that grows exponentially with the sequence length. Here, assuming that all the information the agent needs to act is contained in the current state is impossible [43].

In this paper, we present $\eta\psi$-Learning, an algorithm to learn an exploration policy that methodically searches through a task. This is accomplished by maximizing the entropy of the state visitation distribution of a single finite-length trajectory. This focus on optimizing the state visitation distribution of a single trajectory distinguishes our approach from prior work on exploration methods that maximize the entropy of the state visitation distribution [23, 61, 31, 41, 18]. For example Hazan et al. [23] focus on learning a Markovian policy—a decision-making strategy that is only conditioned

---

[*]Correspondence to <arnav-kumar.jain@mila.quebec>

[†]A part of this work was developed while Lucas Lehnert was at the Mila–Quebec AI Institute and the Université de Montréal.

37th Conference on Neural Information Processing Systems (NeurIPS 2023).

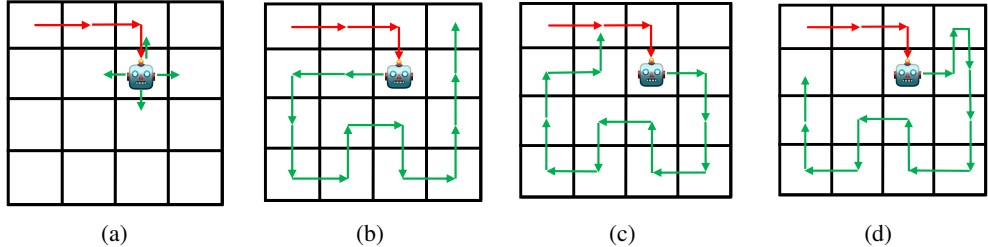

Figure 1: Consider a $4 \times 4$ gridworld for illustration. (a) The agent starts at the top left corner and takes a few actions(red arrows show the trace), (b) an optimal trajectory that covers the grid (green arrows) and can visit all the states without visiting any state twice, (c) a sub-optimal trajectory where agent visits a previously observed state in the last step and not able to visit all the states with the limited steps, (d) another sub-optimal trajectory where agent visits an observed state at an earlier step.

on the current state and does not consider which states have been explored before. A Markovian policy constrains the agent in its ability to express different exploration policies and typically results in randomizing at uncertain states to maximize the state entropy objective. While this will lead to uniformly covering the state space in the limit, such behaviors are not favorable for real-world tasks where the agent needs to maximize state coverage with limited number of interactions.

Figure 1 presents a didactic example to illustrate how an intelligent agent can learn to efficiently explore a $4 \times 4$ grid. In this example, the agent transitions between different grid cells by selecting one of four actions: *up*, *down*, *left*, and *right*. To explore optimally, the agent would select actions that maximize the entropy of the visited state distribution of the entire trajectory. Suppose the agent started its trajectory in the top left corner of the grid (shown in Figure 1(a)) and has moved to the right twice and made one downward step (indicated by red arrows). At this point, the agent has to decide between one of the four actions to further explore the grid. For example, it could move *left* and follow the green trajectory as outlined in Figure 1(b). This path would be optimal in this example because every state is visited exactly once and not multiple times. However, the *top* action would lead to a sub-optimal trajectory as the agent would visit the previous state. To mitigate sub-optimal exploration, an intelligent agent must keep track of visited states to avoid revisiting states. Although the *right* action will lead to a novel state in the next step, the overall behavior will be sub-optimal as the agent will have to visit a state twice to explore the entire grid (depicted in Figures 1(c) and 1(d)). This further requires an agent to carefully plan and account for the states that would follow after taking an action.

In this work, we propose $\eta\psi$-Learning, an algorithm to compute an exploration strategy that methodically explores within a single finite-length trajectory—as illustrated in Figure 1(b). $\eta\psi$-Learning maintains two state representations: a predecessor representation [64, 2] to encode past state visitation frequencies and a Successor Representation (SR) [13] to predict future state visitation frequencies. The two representations are used to evaluate at every time step the decision that leads to covering all states as uniformly as possible. Specifically, for every potential action the agent can take, the SR is combined with the predecessor representation to predict the state visitation distribution for the current trajectory. Then, the action that results in the highest entropy of this state visitation distribution is selected for exploration. Furthermore, this exploration policy can be deterministic and does not randomize to achieve its maximum state entropy objective.

To summarize, the contributions of this work are as follows: Firstly, we propose a mechanism to combine successor [13] and predecessor [64] representations for maximizing the entropy of the state visitation distribution of a finite-length trajectory. To the best of our knowledge, this is the first work using the two representations to optimize the state visitation distribution and learn an efficient exploration policy. Secondly, we introduce $\eta\psi$-Learning, a method that utilizes the combination of two representations to learn deterministic and non-Markovian exploration policies for the finite-sample regime. Thirdly, we discuss how $\eta\psi$-Learning optimizes the entropy-based objective function for both finite and (uncountably) infinite action spaces. In Section 5 we demonstrate through empirical experiments that $\eta\psi$-Learning achieves optimal coverage within a single finite-length trajectory. Moreover, the visualizations presented in Section 5 demonstrate that $\eta\psi$-Learning learns

an exploration policy that maneuvers through the state space to efficiently explore a task while minimizing the number of times the same state is revisited.

## 2 Related Work

The domain of exploration in Reinforcement Learning (RL) focuses on discovering an agent's environment via intrinsic motivation to accelerate learning optimal policies. Many existing exploration methods seek novelty by using prediction errors [45, 9, 54, 57] or pseudo-counts [58, 6, 39]. However, such methods add an intrinsic reward signal to improve sample efficiency in a single task setting. They do not explicitly learn a policy that is designed to efficiently explore a task. In contrast, we present a method for explicitly learning an efficient exploration strategy by maximizing the entropy of a single trajectory's state visitation distribution. Efficient exploration algorithms can be used to improve learning efficiency in application domains such as Meta RL [16, 70, 48, 37], Continual RL [27, 33], and Unsupervised RL [30]. For example, in Meta RL an agent needs to first explore to identify which of the previously observed tasks it is in before the agent can start exploiting rewards in the current task. In this context, VariBAD [70] maintains a belief over which task the agent is in given the observed interactions. While Zintgraf et al. argue that a Bayes-optimal policy implements an efficient exploration strategy, we propose a method that explicitly learns an efficient exploration policy, resulting in discovering rewarding states more efficiently than VariBAD (Section 5).

The core idea behind $\eta\psi$-Learning is the use of the predecessor and successor representations to predict the state visitation distribution induced by a non-Markovian policy for a single finite-length trajectory. Instead of using the successor representation for transfer, lifelong learning, or learning one representation that solve a set of tasks [3, 69, 4, 8, 38, 56, 22, 5, 32, 34, 1, 62], we use the successor representation to estimate the state visitation distribution and maximize its entropy. By using the successor representation in this way, the $\eta\psi$-Learning does not rely on density models [23, 31], an explicit transition model [61, 41], or non-parametric estimators such as k-NN [42]. In the following sections we will discuss how $\eta\psi$-Learning learns a deterministic exploration policy and does not rely on randomization techniques [42, 31] or mixing multiple policies to manipulate the state visitation distribution [31, 23]. Moreover, Mutti et al. [44] provide a theoretical analysis proving that efficient (zero regret) exploration is possible with a deterministic non-Markovian policy but computing such a policy is NP-hard. In this context, $\eta\psi$-Learning is to our knowledge the first algorithm for computing such an efficient exploration policy.

## 3 Maximum state entropy exploration

We formalize the exploration task as a Controlled Markov Process (CMP), a quadruple $\mathcal{M} = \langle \mathcal{S}, \mathcal{A}, p, \mu \rangle$ consisting of a (finite) state space $\mathcal{S}$, a (finite) action space $\mathcal{A}$, a transition function $p$ specifying transition probabilities with $p(s, a, s') = \mathbb{P}(s'|s, a)$, and a start state distribution $\mu$ specifying probability of starting a trajectory at state $s$ with $\mu(s)$. A trajectory is a sequence $\tau = (s_1, a_1, ..., a_{h-1}, s_h)$ of some length $h$ that can be simulated in a CMP. A policy $\pi$ specifies a probability distribution or density function over the action space that is sampled when selecting actions and simulating a trajectory in a CMP. This policy definition include deterministic policies: For discrete action spaces a specific action is selected with probability one and for uncountably infinite action spaces the policy models a delta-dirac density function. In algorithms such as Q-learning [11], the policy is conditioned on the task state and specifies the probabilities of selecting a particular action. However, as illustrated in Figure 1, the past trajectory (red arrows) determines which next action leads to the best exploratory trajectory. Consequently, we consider policies that are functions of trajectories rather than just states.

The state visitation distribution of a trajectory $\tau_h = (s_1, a_1, ..., a_{h-1}, s_h)$ of length $h$ can be formally expressed in a probability vector by first encoding every state $s_t$ as a one-hot bit vector $\boldsymbol{e}_{s_t}$. Using this one-hot encoding, the $h$-step state visitation probability vector of the trajectory $\tau_h$ can computed by marginalizing across the time steps:

$$\boldsymbol{\xi}_{\gamma,\tau} = \sum_{t=1}^{h} \gamma(t)\boldsymbol{e}_{s_t}, \tag{1}$$

where $\gamma : \mathbb{N} \to [0, 1]$ is the *discount function* (we denote the set of positive integers with $\mathbb{N}$), such that $\sum_{t=1}^{h} \gamma(t) = 1$. Using the normalization in the discount function is necessary as it ensures that $\boldsymbol{\xi}_{\gamma,\tau}$ is a probability vector. We note that this use of a discount function is distinct from using a discount factor in common RL algorithms such as Q-learning but using a discount function is necessary as we will elaborate in the following section. The expected state visitation distribution for a policy $\pi$ can be obtained by generating multiple trajectories using $\pi$ and computing the average across them to get the expected state visitation distribution, denoted by $\mathbb{E}_{\tau}[\boldsymbol{\xi}_{\gamma,\tau}]$.

An optimal exploration policy would achieve a similar visitation frequency for each state, as illustrated in Figure 1(b) where the optimal trajectory traverses every state once within the first 15 steps. For this trajectory the vector $\boldsymbol{\xi}_{\tau,\gamma}$ would encode a uniform probability vector, given $\gamma(t) = \frac{1}{h}$ for any $t$. In fact, an optimal exploration policy $\pi^*$ maximizes the entropy of this probability vector and solves the optimization problem

$$\pi^* \in \arg\max_{\pi} H\big(\mathbb{E}_{\tau}\big[\boldsymbol{\xi}_{\gamma,\tau}\big]\big) \tag{2}$$

where the expectation is computed across trajectories that are simulated in a CMP and follow the policy $\pi$.[3] In the remainder of the paper, we will show how optimizing this objective leads to the uniform sweeping behavior illustrated in Figure 1 and the agent learns to maximize the entropy of the state visitation distribution in a single finite length trajectory. In the following section, we describe how $\eta\psi$-Learning optimizes the objective in Equation 2.

## 4    $\eta\psi$-Learning

To learn an efficient exploration policy, we need to estimate the state visitation history and predict the distribution over future states. Consider a trajectory $\tau = (s_1, a_1, ..., s_{T-1}, a_{T-1}, s_T, ...a_{h-1}, s_h)$. At an intermediary step $T$, we denote the $T-1$-step prefix with $\tau_{:T-1} = (s_1, a_1, ..., s_{T-1})$ and the suffix starting at step $T$ with $\tau_{T:} = (s_T, a_T..., a_{h-1}, s_h)$. Using this sub-trajectory notation, the discounted state visitation distribution in Equation 1 can be written as

$$\boldsymbol{\xi}_{\gamma,\tau} = \sum_{t=1}^{T-1} \gamma(t)\boldsymbol{e}_{s_t} + \sum_{t=T}^{h} \gamma(t)\boldsymbol{e}_{s_t}. \tag{3}$$

Assuming the scenario presented in Section 3, suppose the agent has followed the trajectory $\tau_{:T}$ until time step $T$. At this time step, the agent needs to decide which action $a_T$ leads to covering the state space as uniformly as possible and maximizes the entropy of the state visitation distribution. The expected state visitation distribution for a policy $\pi$ can be expressed by conditioning on the trace $\tau_{:T}$ and a potential action $a_T \in \mathcal{A}$:

$$\mathbb{E}_{\tau,\pi}\Big[\boldsymbol{\xi}_{\gamma,\tau}\Big|\tau_{:T}, a_T\Big] = \mathbb{E}_{\tau,\pi}\left[\sum_{t=1}^{T-1} \gamma(t)\boldsymbol{e}_{s_t} + \sum_{t=T}^{h} \gamma(t)\boldsymbol{e}_{s_t}\Big|\tau_{:T}, a_T\right] \tag{4}$$

$$= \underbrace{\sum_{t=1}^{T-1} \gamma(t)\boldsymbol{e}_{s_t}}_{=\boldsymbol{\eta}(\tau_{:T-1})} + \underbrace{\mathbb{E}_{\tau_{T+1:},\pi}\left[\sum_{t=T}^{h} \gamma(t)\boldsymbol{e}_{s_t}\Big|\tau_{:T}, a_T\right]}_{=\boldsymbol{\psi}^{\pi}(\tau_{:T}, a_T)}, \tag{5}$$

where the vector $\boldsymbol{\eta}(\tau_{:T-1})$ is a variant of the predecessor representation [64, 2] and the vector $\boldsymbol{\psi}^{\pi}(\tau_{:T}, a_T)$ is a variant of the successor representation (SR) [13]. Splitting the expected state visitation distribution into a vector $\boldsymbol{\eta}$ and $\boldsymbol{\psi}^{\pi}$ as outlined in Equation 5 is possible because we are assuming a discount function $\gamma$ as defined in Section 3. At time step $T$, the two representations can be added together to estimate the expected state visitation probability vector. Simulating the proposed algorithm is analogous to effectively drawing Monte-Carlo samples from the expectation at different steps $T$ to learn a SR and predict the expected visitation frequencies of $\boldsymbol{\xi}_{\gamma,\tau}$.

The predecessor representation vector $\boldsymbol{\eta}(\tau_{:T-1})$ can still be estimated incrementally similarly to the eligibility trace in TD($\lambda$) algorithm [59] (but with a different weighting scheme that uses the discount function $\gamma$). While the definition of the vector $\boldsymbol{\eta}(\tau_{:T})$ is similar to the definition of eligibility

---

[3]Here, we consider the Shannon entropy $H(\boldsymbol{p}) = -\sum_{i} \boldsymbol{p}_i \log \boldsymbol{p}_i$, where the summation ranges over the entries of the probability vector $\boldsymbol{p}$.

traces [60, Chapter 12], we do not use the predecessor trace for multi-step TD updates to learn more efficiently. Instead, the vector $\boldsymbol{\eta}(\tau_{:T-1})$ estimates the visitation frequencies of past states—the predecessor states—to decide which states to explore next.

While the predecessor representation can be maintained using an update rule because the observed states are known, predicting future state visitation frequencies is more challenging. A potential solution is to exhaustively search through all possible sequences of trajectories starting from the current state. This is computationally infeasible and requires a dynamics model of the environment. Moreover, such a model is not always available, and learning them is prone to errors that compound for longer horizons [49, 26]. To this end, we learn a variant of the successor representation (SR), which predicts the expected frequencies of visiting future or successor states under a policy [13]. In contrast to previous methods which learn successor representation (SR) conditioned on the current state [13, 3], $\eta\psi$-Learning conditions the SR on the entire history of states, given by

$$\boldsymbol{\psi}^\pi(\tau_{:T}, a_T) = \mathbb{E}_{\tau_{T+1:}, \pi} \left[ \sum_{t=T}^{h} \gamma(t) \boldsymbol{e}_{s_t} \middle| \tau_{:T}, a_T \right]. \tag{6}$$

Conditioning the SR on the trajectory $\tau_{:T}$ is necessary because policy $\pi$ is also conditioned on $\tau_{:T}$ and therefore the visitation frequencies of future states depend on $\tau_{:T}$. Moreover, the expectation evaluates all possible trajectories after taking action $a_T$ at time $T$ and following policy $\pi$ afterward. We discuss in Appendix C how the SR vectors are approximated using a recurrent neural network.

We saw in Equation 5 that the predecessor representation and successor representation can be combined to predict the state visitation distribution for a policy $\pi$ and a trajectory-prefix $\tau_{:T}$. $\eta\psi$-Learning uses the estimated state visitation distribution to compute the entropy term in the objective defined in Equation 2. Specifically, the utility function $Q_{\text{expl}}$ approximates the entropy of the state visitation distribution for an action $a_T$ at every time step. By defining

$$Q_{\text{expl}}(\tau_{:T}, a_T) = H\left( \boldsymbol{\eta}(\tau_{:T-1}) + \boldsymbol{\psi}^\pi(\tau_{:T}, a_T) \right), \tag{7}$$

the action that leads to the highest state visitation entropy is assigned the highest utility value. Notably, the proposed Q-function differs from prior methods using the SR, as we neither factorize the reward function [3, 4, 8, 62] nor use the SR for learning a state abstraction [32]. Optimizing the exploration Q-function stated in Equation 7 is challenging as it depends on the SR that itself depends on the policy $\pi$ which changes during learning. Furthermore, Guo et al. [18] show that the Shannon-entropy based objective is difficult to directly optimize using gradient-based methods (due to the log term inside an expectation) [31, 47, 18]. In contrast, we outline in the following paragraphs how the entropy objective in Equation 7 can be directly optimized using either a Q-learning style method or a method based on the Deterministic Policy Gradient [55] framework for finite and infinite action spaces, respectively.

**Finite action space framework** Since the predecessor representation is fixed for a given trajectory $\tau_{:T}$, optimizing the Q-function defined in Equation 7 boils down to predicting the optimal SR for a given history $\tau_{:T}$. Similar to prior Successor Feature learning methods [3, 4, 33], we approximate the SR with a parameterized and differentiable function $\boldsymbol{\psi_\theta}$ and use a loss based on a one-step temporal difference error. Given an approximation $\boldsymbol{\psi_\theta}$, the SR prediction target is obtained by the current state embedding and SR of the optimal action at the next step:

$$\boldsymbol{y}(\tau_{:T+1}, a'_{T+1}) = \boldsymbol{e}_{s_T} + \gamma(T+1) \boldsymbol{\psi_\theta}(\tau_{:T+1}, a'_{T+1}), \tag{8}$$

where $\tau_{:T+1}$ is obtained by adding action $a_T$ and the received next state $s_{T+1}$ to the trajectory $\tau_{:T}$. Analogous to Q-Learning, the optimal action at the next step is specified by

$$a'_{T+1} = \arg\max_{a \in \mathcal{A}} Q_{\text{expl}}(\tau_{:T+1}, a). \tag{9}$$

Being greedy with respect to these entropy values to estimate the target leads to improving the policy $\pi$ which in turn finds the SR for the optimal policy. (Appendix A presents a convergence analysis of this method in a dynamic programming setting.) Then, the function $\boldsymbol{\psi_\theta}$ is optimized using gradient descent on the loss function $\mathcal{L}_{SR}$, given by

$$\mathcal{L}_{SR} = || \boldsymbol{\psi_\theta}(\tau_{:T}, a_T) - \boldsymbol{y}(\tau_{:T+1}, a'_{T+1}) ||^2, \tag{10}$$

where gradients are not propagated through the target $\boldsymbol{y}(\tau_{:T+1}, a'_{T+1})$. Finally, the optimal policy selects actions greedily with respect to the $Q_{\text{expl}}$ function. Algorithm 2 describes the training procedure for the proposed variant for finite action spaces.

**Infinite action space framework** Directly obtaining gradient estimates of objective defined in Equation 2 is challenging because of the expectation term in the non-linear logarithmic term. Previous approaches have used alternative optimization methods [31, 47] or resorted to a simpler noise-contrastive objective function [18]. In contrast with prior algorithms, we derived an $\eta\psi$-Learning variant for infinite action spaces that optimizes an actor-critic architecture using policy gradients to maximise the maximum state entropy objective. The agent uses an actor-critic architecture where actor and critic networks are conditioned on the history of visited states. The actor $\pi_\mu(\tau)$ is parameterized with a parameter vector $\mu$ and is a deterministic map from a trajectory to an action. The critic predicts the SR to estimate the utility function conditioned on a given trajectory and action. Similar to the finite action space variant, the predecessor representation is fixed for a given trajectory and the network has to predict SR $\psi_\theta(\tau, a)$ for a given trajectory $\tau$ and action $a$. Here, the target value of SR to update the critic is specified by the action obtained using the policy $a'_{T+1} = \pi_\mu(\tau_{:T+1})$, given by

$$\boldsymbol{y} = \boldsymbol{e}_{s_T} + \gamma(T+1)\boldsymbol{\psi}_\theta(\tau_{:T+1}, a'_{T+1}). \tag{11}$$

The critic is trained with the same loss function $\mathcal{L}_{SR}$ as defined in Equation 10, where the gradients are not propagated through the target. The actor is optimized to maximize the estimated utility function (Equation 2). Since the actor is deterministic, policy gradients are computed using an adaptation of the deterministic policy gradient theorem [55]. Because the actor network has no dependency on predecessor trace (which depends on the observed states only), gradients for the actor parameters are obtained by applying chain rule leading to the following gradient of Equation 7 (please refer to Proposition 2 for more details on the derivation):

$$\nabla_\mu J(\pi_\mu) = \mathbb{E}_{\tau \sim \rho}\Big[\sum_i z_i \nabla_\mu \pi_\mu(\tau) \nabla_a \boldsymbol{\psi}_i(\tau, a)\big|_{a=\pi_\mu(\tau)}\Big], \tag{12}$$

where $z_i = -\log[\boldsymbol{\eta}(\tau_{:-1})_i + \boldsymbol{\psi}(\tau, \pi_\mu(\tau))_i] - 1$ is the multiplicative factor for state $i$, and depends on the expected probability of visiting a state. The factor $z_i$ can take values between between -1 and $\infty$, with positive values of high magnitude for states with low visitation probability and negative values for state with high probability of visitation. Thus, the factor $z_i$ scales the policy gradients to maximize the entropy of the state visitation distribution. In Algorithm 3 we outline the training procedure for the infinite action space framework.

## 5   Experiments

To analyze and test if $\eta\psi$-Learning learns an efficient exploration policy, we evaluate the proposed method on a set of discrete and continuous control tasks. In these experiments, we are recording a set of different performance measures to access if the resulting exploration policies do in fact maximize the entropy of the state visitation distribution and if most states are explored by $\eta\psi$-Learning. The following results demonstrate that by maintaining a predecessor representation and conditioning the SR on the simulated trajectory prefix, the $\eta\psi$-Learning agent learns a deterministic exploration policy that minimizes the number of interactions needed to visit all states. In addition, we expand our method to continuous control tasks and demonstrate how $\eta\psi$-Learning can efficiently explore in complex domains with infinite action space. Our method is ideal for searching out rewards in difficult sparse reward environments. We compare $\eta\psi$-Learning, which learns to explore an environment as efficiently as possible, to recent meta-RL methods [70] that aim to learn how to optimally explore an environment to infer the rewarding or goal state. Lastly, we show the applicability of $\eta\psi$-Learning on standard RL tasks where the extrinsic rewards are sparse and present how the proposed method can be combined with existing algorithms.

**Environments**: We experiment with different tasks with both finite and infinite action spaces. The **ChainMDP** and **RiverSwim** [58] is a six-state chain where the transitions are deterministic or stochastic, respectively. In these tasks a Markovian policy cannot cover the state space uniformly because the agent has to pace back and forth along the chain, visiting the same state multiple times. For the RiverSwim environment , a non-stationary policy, a policy that is a function of the time step, cannot optimally cover all states because non-determinism in the transitions can place the agent into different states at random. Furthermore, we include the $5 \times 5$ **grid world** example used in Figure 1. We also test $\eta\psi$-Learning on two harder exploration tasks—the **TwoRooms** and **FourRooms** domains, which are challenging because it is not possible to obtain exact uniform visitation distribution due to the wall structure. For continuous control tasks, we evaluate on Reacher and Pusher tasks, where the

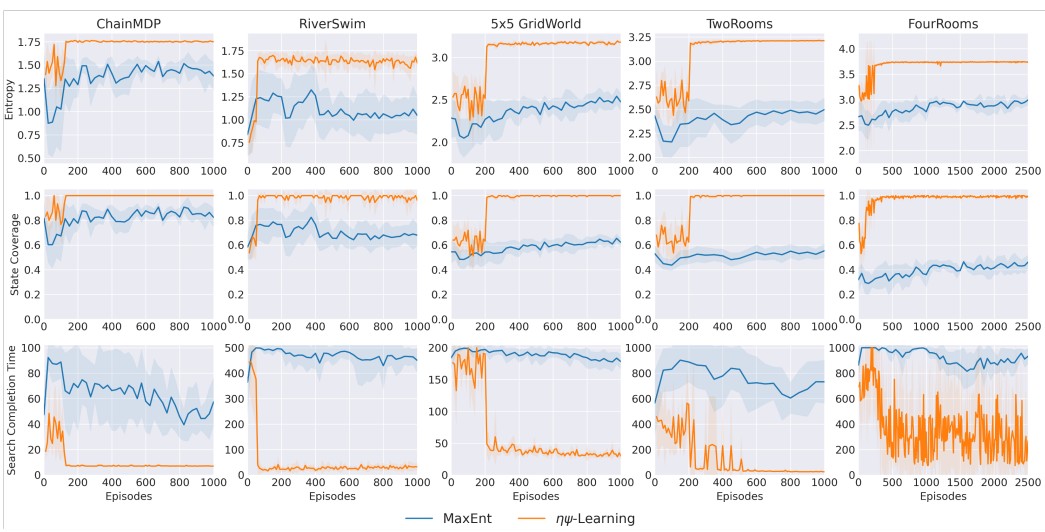

Figure 2: Comparison of $\eta\psi$-Learning and MaxEnt [23] on three metrics: *Entropy* (top row) of state visitation distribution, *State Coverage* (middle row) representing the fraction of state space visited, and *Search Completion Time* (bottom row) denoting steps taken to cover the state space.

agent has a robotic-arm with multiple joints. The task is to maximize the entropy over the locations covered by the fingertip of the robotic-arm. Appendix F provides more details on the environments and the hyper-parameters are reported in Appendix G.

**Prior Methods**: To our knowledge, existing work focusses on learning Markovian exploration policies [43]. We use MaxEnt [23] as a baseline agent for our experiments because this method optimizes a similar maximum entropy objective as $\eta\psi$-Learning—with the difference that MaxEnt learns a Markovian policy and resorts to randomization to obtain a close to uniform state visitation distribution. We have also compared with MEPOL [42] on continuous control tasks which leans a Markovian policy and uses kNN-based estimators to compute the entropy. A comparison with SMM [31] is skipped because the method optimizes a similar maximum entropy objective with a Markovian policy and cannot express the same exploration behaviour as $\eta\psi$-Learning.

**Evaluation Metrics**: *Entropy* measures a method's ability to have similar visitation frequency for each state in the state space. This signifies the gap between the observed state visitation distribution and the optimal distribution that maximizes the entropy term. The Entropy metric is computed using the objective defined in Equation 2 over a single trajectory generated by the agent. A constant discount factor of $\gamma(t) = \frac{1}{h}$ is used to obtain the state visitation distribution during evaluation. An agent can maximize this measure without actually exploring all states of an environment—a desirable property for RL where rewards may be sparse and hidden in complex to-reach states. We record the *state coverage* metric which represents the fraction of states in the environment visited by the agent at least once within a trajectory. Lastly, we want agents to explore the state space efficiently. For example, an optimal agent can sweep through the gridworld presented in Figure 1 with a search completion time of 15 steps ( Figure 1(b) shows an optimal trajectory). The *search completion time* metric measures the steps taken to discover each state in the environment. All results report the mean performance computed over 5 random seeds with 95% confidence intervals shading.

**Quantitative Results**: Figure 2 presents the results obtained for $\eta\psi$-Learning and MaxEnt [23]. Compared to MaxEnt, which learns a Markovian policy, $\eta\psi$-Learning achieves 20-50% higher entropy. This indicates that the MaxEnt algorithm by learning a Markovian and stochastic policy was randomizing at certain states which lead to sub-optimal behaviors. The performance gain was more prominent in grid-based environments because the MaxEnt agent was visiting some states more frequently than others, which are harder to explore efficiently. Furthermore, high *entropy* values suggest that the agent visits states with similar frequency in the environment and does not get stuck at a particular state. We attribute this behavior of $\eta\psi$-Learning to the proposed Q-function that picks action to visit different states and maximize the objective.

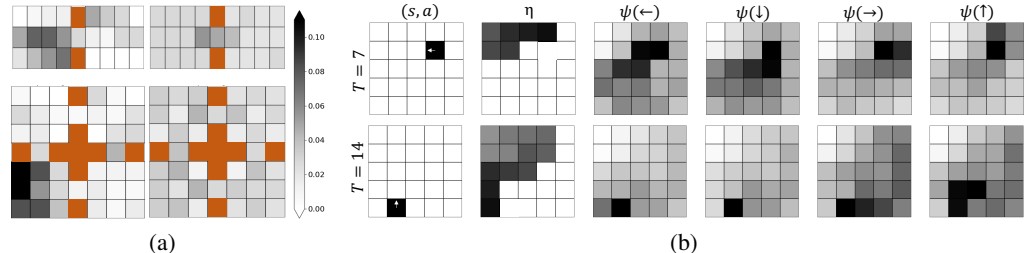

Figure 3: (a) Heatmap of state visitation distribution by unrolling a trajectory using MaxEnt (left) and $\eta\psi$-Learning (right) on TwoRooms and FourRooms environments. (b) Visualization of learned SR of each action (denoted with $\psi(.)$) at time steps $T = 7, 14$ for a trajectory using $\eta\psi$-Learning on $5 \times 5$ grid. $(s, a)$ denotes the state (black) and action taken by the agent (direction of white arrow), $\boldsymbol{\eta}$ is the predecessor representation till time $T$ (higher values have darker shade)

.

Figure 2 also shows that $\eta\psi$-Learning achieves optimal state coverage across environments exemplifying that $\eta\psi$-Learning while maximizing the entropic measure also learns to cover the state space within a single trajectory. However, the baseline MaxEnt was not able to discover all the states in the environment. MaxEnt was unable to visit all the states in ChainMDP and RiverSwim environments with trajectory length of 20 and 50, respectively. Moreover, the state coverage of MaxEnt was around **50-60%** on the harder TwoRooms and FourRooms tasks, where the agent has to navigate between different rooms and is required to remember the order of visiting different rooms. These results reveal that Markovian policy limits an agent's ability to maximize the state coverage in a task. The proposed method also outperformed the baseline on the *search completion time* metric across all environments. Notably, on ChainMDP, the $5 \times 5$ gridworld, and TwoRooms environments, $\eta\psi$-Learning converged within 500 episodes. However, $\eta\psi$-Learning did not achieve optimal *search exploration time* on the FourRooms environment as it missed a spot in a room and resorted to it later in the episode.

To further understand the gains offered by $\eta\psi$-Learning, we visualized the state visitation distributions on a single trajectory (in Figure 3(a)). On the TwoRooms environment, $\eta\psi$-Learning had similar density on the states in both the rooms, where the density is more around the center. This is because the agent was sweeping across rooms alternatively. $\eta\psi$-Learning showed a better-visited state distribution on the FourRooms environment with more distributed density across states. However, MaxEnt was not visiting all states and also visited a few states more frequently than others, elucidating the lower performance on entropy and state coverage. We further visualized the learned SR to see if $\eta\psi$-Learning learns a SR for the optimal exploration policy through generalized policy improvement [3]. For this analysis, we sampled a trajectory on $5 \times 5$ gridworld. Figure 3(b) reports the heatmaps of the learned SR vector for each action at different steps in the trajectory. We observe that the SR vector for each action has lower density on the states already observed in the trace. This exemplifies that the learned SR captures the history of the visited states that further aids in taking actions to maximize the entropy of state visitation distribution. We also study if MaxEnt can show similar gains when trained with a recurrent policy (Appendix I.1) and compared the agents when evaluated across multiple trajectories (Appendix I.2).

**Continuous Control tasks**: The efficacy of $\eta\psi$-Learning is further tested on environments with infinite action space. Figure 4(a) reports the Entropy and State Coverage metric on Reacher and Pusher environments, where $\eta\psi$-Learning outperformed the baseline MaxEnt on both metrics. The gains are more significant on the Pusher environment which is a harder task because of multiple hinges in the robotic-arm. The proposed method $\eta\psi$-Learning achieves close to **90%** coverage in both environments, whereas the MaxEnt had only close to **50%** and **40%** coverage on Reacher and Pusher environments, respectively. In Figure 4(b), heatmaps of the state visitation density for a single trajectory shows that $\eta\psi$-Learning has more uniformly distributed density compared to MaxEnt. The Pusher environment has high density at the top-right corner of the grid denoting the time taken by the agent to move the fingertip to other locations from the starting state. Notably, the proposed method $\eta\psi$-Learning has lower density at the starting state and we believe that conditioning on the history of states is guiding the agent to move the robotic-arm to other locations to maximize the entropy over the state space. In Appendix I.5, a visualization of a rolled-out trajectory generated using $\eta\psi$-Learning is

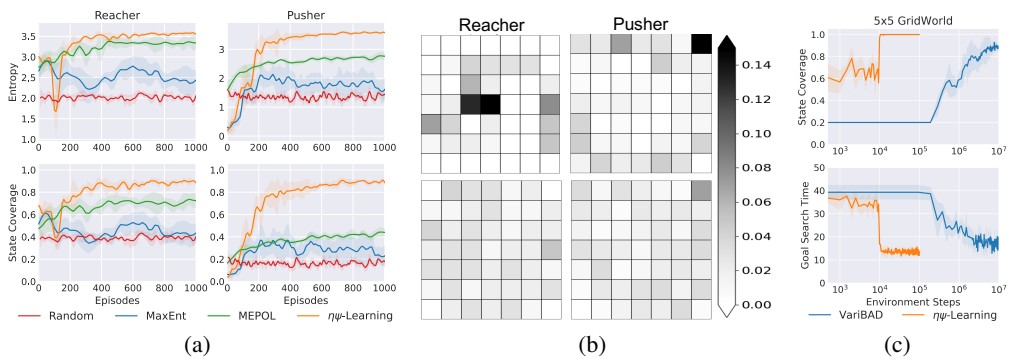

Figure 4: (a) Comparison of $\eta\psi$-Learning with Random policy, MaxEnt [43], and MEPOL [42] on Reacher and Pusher tasks (b) Heatmaps of visitation distribution of MaxEnt (top) and $\eta\psi$-Learning (bottom), (c) comparison with VariBAD [70] on State Coverage and Goal Search Time metrics.

presented showing that the agent learns to efficiently maneuver the fingertip of the robotic-arm to different locations in the environment.

**Comparison with Meta-RL**: A question central to Meta-RL [16, 70, 37] is the ability to quickly explore a task and find rewarding states in complex tasks where the rewards are sparse. In this context, Zintgraf et al. [70] present the VariBAD method, which maintains a belief over different tasks to infer the optimal policy—leading to efficient exploration behaviour that enables the agent to discover rewarding states quickly. Similar to VariBAD, the predecessor representation $\boldsymbol{\eta}$ in $\eta\psi$-Learning keeps track of which states have been explored and which states are not explored. Figure 4(c) compares the exploration behaviour of $\eta\psi$-Learning to VariBAD: In terms of the State Coverage and Goal Search Time metric, $\eta\psi$-Learning outperforms VariBAD significantly because $\eta\psi$-Learning is designed to optimize the entropy of the state visitation frequencies of a single trajectory instead of performing Bayes-adaptive inference across a task space. We refer the reader to Appendix H for more details on this experiment.

$\eta\psi$-**Learning in Sparse Reward Tasks**: Sparse reward environments pose a challenge where an agent has to discover the reward function by visiting different regions of the state space. Through this experiment, we demonstrate how an agent can leverage $\eta\psi$-Learning as an exploratory bonus and improve its efficiency in such tasks. The experiment is conducted on Sparse Mountain-Car environment where the agent receives a positive reward upon reaching the goal position. This is hard because the agent needs to plan to visit different positions in the environment. We have used TD3 [17] as the baseline algorithm for this task. To compare with leading exploration methods, we add TD3 combined with Count-Based [6] and First-Occupancy based [40] to the baselines. It is to be noted that we have used these methods as episodic bonuses as described in [67]. We also propose a variant of TD3 equipped with the proposed method and call it TD3-$\eta\psi$-Learning, where we learn two critics to estimate the sum of extrinsic rewards and the entropy of state visitation distribution (Equation 7). Appendix J describes the experimental setup in more detail and Algorithm 4 provides the pseudocode of the proposed variant. The evaluation is done across two metrics- Average Return and Average Episode Length denoting the steps taken to reach the goal state. Figure 5 presents the results where the proposed method outperforms the baseline algorithms and is sample-efficient at learning the task.

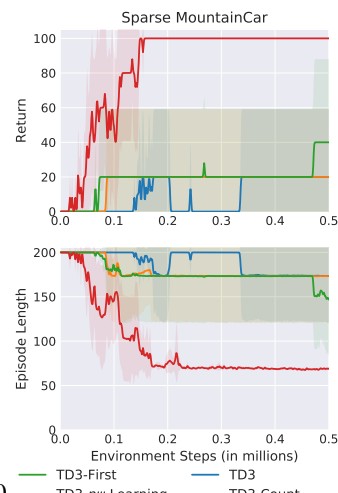

Figure 5: Results on Sparse Mountain Car environment.

TD3 combined with First-Occupancy based intrinsic reward performed better than Count-Based bonus but still takes twice number of steps as the proposed method to solve the task. This shows that the proposed method can improve sample efficiency in standard RL tasks, especially with sparse

rewards. We leave scaling the proposed method with leading architectures [21, 25, 53] to more challenging tasks for future research.

## 6 Discussion

To explore efficiently, an intelligent agent needs to consider past episodic experiences to decide on the next exploratory action. We demonstrate how the predecessor representation—an encoding of past state visitations—can be combined with the successor representation—a prediction of future state visitations—to learn efficient exploration policies that maximize the state-visitation-distribution entropy. Across a set of different environments, we illustrate how $\eta\psi$-Learning consistently reasons across different trajectories to explore near optimally—a task that is NP-hard [43].

To the best of our knowledge, $\eta\psi$-Learning is the first algorithm that combines predecessor and successor representations to estimate the state visitation distribution. Furthermore, $\eta\psi$-Learning learns a non-Markovian policy and can therefore express exploration behavior not afforded by existing methods [23, 31, 42, 18]. To further increase the applicability of $\eta\psi$-Learning, one interesting direction of future research is to extend $\eta\psi$-Learning to POMDP environments where states are either partially observable or complex such as images. This is challenging because the agent has to learn an embedding of state observations that capture only the relevant components of the state space to maximize the entropy. We believe a promising approach would be to leverage the idea of Successor Measures (SM) [62, 63, 15] which have shown promising results when scaled to high-dimensional inputs like images. Furthermore, the presented approach can be also used for designing other algorithms that control the state visitation distribution. An application is goal-conditioned RL, where the agents need to minimize the KL divergence between visitation distribution of policy and goal-distribution [31, 47]. Another application is Safe RL [66] where agents receive a penalty upon visiting unsafe states to avoid observing them.

We study reinforcement learning, which aims to enable autonomous agents to acquire search behaviors. This study of developing exploration behaviors in reinforcement learning is guided by a fundamental curiosity about the nature of autonomous learning; it has a number of potential practical applications and broad implications. First, autonomous exploration for pre-training in general, can enable autonomous agents to acquire useful skills with less human intervention and effort, potentially improving the feasibility of learning-enabled robotic systems. Second, the practical applications that we illustrate, such as applications to continuous environments, can accelerate reinforcement learning in certain settings. Specific to our method, finite length entropy maximization may also in the future offer a useful tool for search and rescue, by equipping agents with an objective that causes them to explore a space systematically to locate lost items. However, these types of reinforcement learning methods also have a number of uncertain broad implications: agents that explore the environment and attempt to acquire open-ended skills may carry out unexpected or unwanted behaviors, and would require suitable safety mechanisms of their own during training.

## Acknowledgements

The authors would like to thank Harley Wiltzer for his valuable feedback and discussions. The writing of the paper also benefited from discussions with Darshan Patil, Chen Sun, Mandana Samiei, Vineet Jain, and Arushi Jain. The authors want to acknowledge funding support from NSERC and CIFAR, AJ and IR acknowledge the support from Canada CIFAR AI Chair Program and from the Canada Excellence Research Chairs (CERC) program. The authors are also grateful to Mila (mila.quebec) IDT, NVIDIA and Digital Research Alliance of Canada for computing resources.

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

# A Convergence Analysis

To gain a deeper understanding why the $\eta\psi$-Learning converges to a maximum entropy policy, we consider in this section a simplified dynamic programming variant in Algorithm 1. Note that the $\eta\psi$-Learning estimates the SR for a finite CMP for a finite horizon length $h$. Consequently, the trajectory-action conditioned SR $\psi^\pi(\tau_{:T}, a)$ and exploration policy $\pi$ can be stored in an exponentially large but finite look-up table. Furthermore, with every transition an additional state is appended to the trajectory $\tau_{:T}$, meaning the agent cannot loop back to the same trajectory. Using these two properties, we state a dynamic programming variant of $\eta\psi$-Learning in Algorithm 1 and then prove its convergence to a policy that maximizes the entropy term $H\left(\boldsymbol{\eta}(\tau_{:T-1}) + \boldsymbol{\psi}^\pi(\tau_{:T}, a)\right)$ at every time step.

---

**Algorithm 1** $\eta\psi$-Learning: Dynamic Programming Framework

---

1: **for all** $\tau_h, a$ **do**
2:   $\boldsymbol{\psi}^\pi(\tau_{:h}, a) \leftarrow \boldsymbol{e}_{s_h}$
3: **end for**
4: **for** $t = h, ..., 2$ **do**
5:   **for all** $\tau_{:t}, a$ **do**
6:     $\pi(\tau_{:t}) \leftarrow \arg\max_a H\left(\boldsymbol{\eta}(\tau_{:t-1}) + \boldsymbol{\psi}^\pi(\tau_{:t}, a)\right)$
7:     $\boldsymbol{\psi}^\pi(\tau_{:t-1}, a) \leftarrow \boldsymbol{e}_{s_{t-1}} + \gamma(t)\boldsymbol{\psi}^\pi(\tau_{:t}, \pi(\tau_{:t}))$
8:   **end for**
9: **end for**
10: **return** $\pi$ such that $\pi(\tau) = \arg\max_a H(\boldsymbol{\eta}(\tau_{:-1}) + \boldsymbol{\psi}(\tau, a))$.

---

The convergence proof uses the following property of the predecessor trace $\boldsymbol{\eta}$ and SR $\boldsymbol{\psi}^\pi$: Consider a trajectory $\tau$ which selects action $a_T$ at time step $T$, then

$$\boldsymbol{\eta}(\tau_{:T-1}) + \boldsymbol{\psi}^\pi(\tau_{:T}, a_T) = \underbrace{\sum_{t=1}^{T-1} \gamma(t)\boldsymbol{e}_{s_t}}_{=\boldsymbol{\eta}(\tau_{:T-1})} + \underbrace{\mathbb{E}_{\tau_{T+1:},\pi}\left[\sum_{t=T}^{h} \gamma(t)\boldsymbol{e}_{s_t} \middle| \tau_{:T}, a_T\right]}_{=\boldsymbol{\psi}^\pi(\tau_{:T}, a_T)} \qquad \text{(by Eq. 5)}$$

$$= \sum_{t=1}^{T-1} \gamma(t)\boldsymbol{e}_{s_t} + \mathbb{E}_{\tau_{T+1:},\pi}\left[\gamma(T)\boldsymbol{e}_{s_T} + \sum_{t=T+1}^{h} \gamma(t)\boldsymbol{e}_{s_t} \middle| \tau_{:T}, a_T\right] \qquad (13)$$

$$= \sum_{t=1}^{T} \gamma(t)\boldsymbol{e}_{s_t} + \mathbb{E}_{\tau_{T+1:},\pi}\left[\sum_{t=T+1}^{h} \gamma(t)\boldsymbol{e}_{s_t} \middle| \tau_{:T}, a_T\right] \qquad (14)$$

$$= \boldsymbol{\eta}(\tau_{:T}) + \mathbb{E}_{\tau_{T+1:},\pi}\left[\boldsymbol{\psi}^\pi(\tau_{:T+1}, \pi(\tau_{:T+1})) | \tau_{:T}, a_T\right] \qquad (15)$$

Using this identity, we can prove the convergence of Algorithm 1.

**Proposition 1.** The policy $\pi^*$ returned by Algorithm 1 is such that for every $t$-step trajectory $\tau_{:t}$ where $t \leq h$,

$$\pi^*(\tau_{:t}) \in \arg\max_\pi H(\boldsymbol{\eta}(\tau_{:t-1}) + \boldsymbol{\psi}^\pi(\tau_{:t}, \pi(\tau_{:t}))). \qquad (16)$$

*Proof.* The proof proceeds by induction on the length of an $h$-step trajectory, starting with a length of $h$ and iterating to a length of one.

**Induction hypothesis:** We define a sub-sequence optimal policy $\pi_t$ such that for every $k$-step trajectory prefix $\tau_{:k}$ and $t \leq k \leq h$,

$$\pi_t \in \arg\max_\pi H(\boldsymbol{\eta}(\tau_{:k-1}) + \boldsymbol{\psi}^\pi(\tau_{:k}, \pi(\tau_{:k}))). \qquad (17)$$

The exploration policy $\pi_t$ is the optimal after executing the first $t$ steps of an $h$-step trajectory $\tau$. The goal is to prove that the induction hypothesis in line (17) holds for $t = 1$.

**Base case:** The base case for $t = h$ holds trivially, because SR does not have a dependency on the policy $\pi$ for an $h$-step trajectory. Therefore the policy $\pi$ can output any action for a trajectory sequence of length $h$:

$$\max_\pi H(\boldsymbol{\eta}(\tau_{:h-1}) + \boldsymbol{\psi}^\pi(\tau_{:h}, \pi(\tau_{:h}))) = H(\boldsymbol{\eta}(\tau_{:h-1}) + \gamma(h)\boldsymbol{e}_{s_h}). \qquad (18)$$

**Induction Step:** Suppose the induction hypothesis in line (17) holds for some $t > 1$ and $\pi_t$ is the maximizer of

$$H(\boldsymbol{\eta}(\tau_{:k-1}) + \boldsymbol{\psi}^{\pi_t}(\tau_{:k}, \pi_t(\tau_{:k}))) \tag{19}$$

where $t \leq k \leq h$. For time step $t-1$, we have that for some action $a$,

$$H(\boldsymbol{\eta}(\tau_{:t-2}) + \boldsymbol{\psi}^{\pi_t}(\tau_{:t-1}, a)) = H(\boldsymbol{\eta}(\tau_{:t-2}) + \gamma(t-1)\boldsymbol{e}_{s_{t-1}} + \mathbb{E}\left[\boldsymbol{\psi}^{\pi_t}(\tau_{:t}, \pi_t(\tau_{:t})|s_{t-1}, a]\right) \tag{20}$$

$$= H(\boldsymbol{\eta}(\tau_{:t-1}) + \mathbb{E}\left[\boldsymbol{\psi}^{\pi_t}(\tau_{:t}, \pi_t(\tau_{:t}))|s_{t-1}, a]\right) \tag{21}$$

$$= H(\mathbb{E}\left[\boldsymbol{\eta}(\tau_{:t-1}) + \boldsymbol{\psi}^{\pi_t}(\tau_{:t}, \pi_t(\tau_{:t}))|s_{t-1}, a]\right). \tag{22}$$

We note that the term inside the expectation is already maximized by $\pi_t$ (by induction hypothesis). If we now set $\pi_{t-1}$ to be equal to $\pi_t$ for every $t$-step or longer trajectory and set

$$\pi_{t-1}(\tau_{:t-1}) = \arg\max_a H(\boldsymbol{\eta}(\tau_{:t-2}) + \boldsymbol{\psi}^{\pi_t}(\tau_{:t-1}, a)), \tag{23}$$

then for $t - 1 \leq k \leq h$

$$\pi_{t-1} \in \arg\max_\pi H(\boldsymbol{\eta}(\tau_{:k-1}) + \boldsymbol{\psi}^\pi(\tau_{:k}, \pi(\tau_{:k}))). \tag{24}$$

This completes the proof. $\qquad\square$

# B $\eta\psi$-Learning- Policy Gradient

Application of Q-Learning based approaches to continuous action space is not easy because finding the greedy action at any time step can be slow to be practical with large, unconstrained function approximators and nontrivial action spaces. In this work, we take a similar approach to deterministic policy gradient [55] to learn exploratory policies. The objective remains the same which is to maximize the entropy of state visitation distribution. However, it is challenging to estimate the gradient where the objective is based on the entropy term. Previous works have either used alternate optimization [47, 31] or similar objective functions [18]. The challenge is because of the expectation inside the logarithm in Equation 2. [31, 47] addressed this intractability by first estimating the visited state distribution and then using this estimate to optimize the entropy-based objective. Unfortunately, such alternating approaches are often are prone to instability and slow convergence [18]. In this work, we take an alternative direction and learn a network to directly estimate the visited state distribution. The combination of predecessor trace $\boldsymbol{\eta}$ and successor representation $\boldsymbol{\psi}^\pi$ can be leveraged to estimate the state visitation distribution which is obtained using:

$$\boldsymbol{\eta}(\tau_{:T-1}) + \boldsymbol{\psi}^\pi(\tau_{:T}, a_T) = \underbrace{\sum_{t=1}^{T-1} \gamma(t)\boldsymbol{e}_{s_t}}_{=\boldsymbol{\eta}(\tau_{:T-1})} + \underbrace{\mathbb{E}_{\tau_{T+1:}, \pi}\left[\sum_{t=T}^h \gamma(t)\boldsymbol{e}_{s_t} \middle| \tau_{:T}, a_T\right]}_{=\boldsymbol{\psi}^\pi(\tau_{:T}, a_T)} \quad \text{(by Eq. 5)} \tag{25}$$

The SR vector can be learned with gradient based optimization and provides the estimate of state visitation distribution for a given policy $\pi$ and trajectory. The policy can utilize this estimate to learn optimal behaviors for efficient exploration in the environment.

To learn optimal behaviors for continuous action spaces, $\eta\psi$-Learning uses an actor-critic architecture comprising of a deterministic actor $\pi_\mu(\tau)$ that provides the action and a critic to estimate the utility function. Here, both the actor and critic networks are non-Markovian and depend on the entire history of visited states. The goal of the critic network is to approximate the Q-function for a given trajectory $\tau$ and a given action $a \in \mathcal{A}$. For a given trajectory $\tau_{:T}$, critic computes this by combining the predecessor representation and the SR vector. The predecessor representation is fixed for a given history, implying that the critic only needs to approximate the SR $\boldsymbol{\psi}_\theta(\tau, a)$. To summarize, the critic estimates the Q-function as shown below:

$$Q_{\theta,\text{expl}}(\tau_{:T}, a_T) = H\left(\boldsymbol{\eta}(\tau_{:T-1}) + \boldsymbol{\psi}_\theta(\tau_{:T}, a_T)\right), \tag{26}$$

To update the critic network, we update the SR approximator network using temporal-difference error. The target for the SR is obtained using the action coming from the current policy $a'_{T+1} = \pi_\mu(\tau_{:T+1})$, and is given by

$$\boldsymbol{y} = \boldsymbol{e}_{s_T} + \gamma(T+1)\boldsymbol{\psi}_\theta(\tau_{:T+1}, a'_{T+1}). \tag{27}$$

The SR network is updated with gradient-based learning to optimize the Mean-Squared Error between the predicted SR and the target, and the loss function $\mathcal{L}_{SR}$ is given by

$$\mathcal{L}_{SR} = ||\boldsymbol{\psi_\theta}(\tau_{:T}, a_T) - \boldsymbol{y}(\tau_{:T+1}, a'_{T+1}).||^2 \qquad \text{(by Eq. 10)}$$

Given an estimate of the SR for the current policy, we need a mechanism to update the actor network to maximize the objective. Deterministic policy gradient algorithm [55] provided a way of learning optimal policies with a deterministic actor. In this work, we formulate the gradient for the actor parameters using similar mechanism with the goal to maximize the entropy-based utility function. Proposition 2 presents a derivation of the gradients for the actor network parameters obtained by applying the chain rule on the Shannon-entropy based Q-function.

**Proposition 2.** Assuming the CMP satisfies [55, conditions A.1] (all functions are continuous and differentiable across all parameters) and for a $\mu$-parameterized policy function $\pi_\mu$ the gradient with respect to $\mu$ of the maximum entropy objective

$$J(\pi_\mu) = \mathbb{E}_{\tau \sim \rho}[H(\boldsymbol{\eta}(\tau_{:-1}) + \boldsymbol{\psi}(\tau, \pi_\mu(\tau)))]$$

is

$$\nabla_\mu J(\pi_\mu) = \mathbb{E}_{\tau \sim \rho}\Big[\sum_i z_i \nabla_\mu \pi_\mu(\tau) \nabla_a \boldsymbol{\psi}_i(\tau, a)\big|_{a = \pi_\mu(\tau)}\Big].$$

where $z_i = -\log[\boldsymbol{\eta}(\tau_{:-1})_i + \boldsymbol{\psi}(\tau, \pi_\mu(\tau))_i] - 1$, $H$ is the Shannon-Entropy function over the representation vectors, and the expectation over trajectories is computed with respect to some trajectory visitation distribution $\rho$.

*Proof.* We begin by rewriting the Shannon Entropy here for a $T$-step trajectory as

$$H(\boldsymbol{\eta}(\tau_{:T-1}) + \boldsymbol{\psi}(\tau_{:T}, a_T)) = -\sum_i (\boldsymbol{\eta}(\tau_{:T-1})_i + \boldsymbol{\psi}(\tau_{:T}, a_T)_i) \log((\boldsymbol{\eta}(\tau_{:T-1})_i + \boldsymbol{\psi}(\tau_{:T}, a_T)_i),$$

$$(28)$$

where $a_T = \pi_\mu(\tau_{:T})$.

To simplify the notations, we will use $\boldsymbol{\eta}_i = \boldsymbol{\eta}(\tau_{:T-1})_i$ and $\boldsymbol{\psi}_i = \boldsymbol{\psi}(\tau_{:T}, a_T)_i$ to represent the $i$th term of the predecessor and successor representation vectors. Now taking the gradient with respect to the actor parameters $\mu$ gives:

$$\nabla_\mu H(\boldsymbol{\eta}_i + \boldsymbol{\psi}_i) = -\nabla_\mu \sum_i (\boldsymbol{\eta}_i + \boldsymbol{\psi}_i) \log(\boldsymbol{\eta}_i + \boldsymbol{\psi}_i) \tag{29}$$

$$= -\sum_i [\nabla_\mu (\boldsymbol{\eta}_i + \boldsymbol{\psi}_i) \log(\boldsymbol{\eta}_i + \boldsymbol{\psi}_i)] \tag{30}$$

$$= -\sum_i [\log(\boldsymbol{\eta}_i + \boldsymbol{\psi}_i) \nabla_\mu (\boldsymbol{\eta}_i + \boldsymbol{\psi}_i) + (\boldsymbol{\eta}_i + \boldsymbol{\psi}_i) \nabla_\mu \log(\boldsymbol{\eta}_i + \boldsymbol{\psi}_i)] \tag{31}$$

$$= -\sum_i [\log(\boldsymbol{\eta}_i + \boldsymbol{\psi}_i) \nabla_\mu \boldsymbol{\psi}_i + \nabla_\mu \boldsymbol{\psi}_i \tag{32}$$

$$= -\sum_i [\log(\boldsymbol{\eta}_i + \boldsymbol{\psi}_i) + 1] \nabla_\mu \boldsymbol{\psi}_i \tag{33}$$

Now, using the chain rule on the $i$th feature in SR, we obtain

$$\nabla_\mu \boldsymbol{\psi}_i = \nabla_\mu \pi_\mu(\tau_{:T}) \nabla_a \boldsymbol{\psi}(\tau_{:T}, a)_i\big|_{a = \pi_\mu(\tau_{:T})} \tag{34}$$

By substitution

$$\nabla_\mu H(\boldsymbol{\eta}_i + \boldsymbol{\psi}_i) = -\sum_i [\log(\boldsymbol{\eta}_i + \boldsymbol{\psi}_i) + 1] \nabla_\mu \pi_\mu(\tau_{:T}) \nabla_a \boldsymbol{\psi}(\tau_{:T}, a)_i\big|_{a = \pi_\mu(\tau_{:T})}. \tag{35}$$

Therefore, the gradient of the overall objective is

$$\nabla_\mu J(\pi_\mu) = \mathbb{E}_{\tau \sim \rho}[\nabla_\mu H(\boldsymbol{\eta}(\tau_{:-1}) + \boldsymbol{\psi}(\tau, \pi_\mu(\tau)))] \tag{36}$$

$$= -\mathbb{E}_{\tau \sim \rho}\left[\sum_i [\log(\boldsymbol{\eta}_i + \boldsymbol{\psi}_i) + 1] \nabla_\mu \pi_\mu(\tau_{:T}) \nabla_a \boldsymbol{\psi}(\tau_{:T}, a)_i\big|_{a = \pi_\mu(\tau_{:T})}\right]. \tag{37}$$

This completes the proof. $\qquad\qquad\square$

**Algorithm 2** $\eta\psi$-Learning: Finite Action Space Framework

---

 1: Initialize SR network with parameters $\theta$ and the replay buffer $\mathcal{B} = \{\}$
 2: Denote the predecessor feature with $\boldsymbol{\eta}$, discount function with $\gamma$, and episode length with $h$
 3: **while** Training **do**
 4:     Collect $\tau_{exp} = \{s_1, a_1, .., s_h\}$ using current policy $\pi_\theta$ and add it to replay buffer $\mathcal{B} = \mathcal{B} \cup \tau_{exp}$
 5:     **for** each training step **do**
 6:         Sample batch of $\tau = (s_1, .. a_{l-1}, s_l) \sim \mathcal{B}$ of sequence length $l \in \{2, .., h\}$
 7:         Compute $a' = \arg\max_{a \in \mathcal{A}} H(\boldsymbol{\eta}(\tau) + \boldsymbol{\psi}_\theta(\tau, a))$
 8:         Compute target $\boldsymbol{y} = \boldsymbol{e}_{s_{l-1}} + \gamma(l)\boldsymbol{\psi}_\theta(\tau, a')$
 9:         Update SR network by performing gradient step on $||\boldsymbol{y} - \boldsymbol{\psi}_\theta(\tau_{:l-1}, a_{l-1})||_2^2$
10:     **end for**
11: **end while**

---

In Proposition 2, we derive the gradient of the actor parameters for the maximum state entropy exploration objective. Taking inspiration from algorithms [35, 17, 19] that extend Deterministic Policy Gradient (DPG) to make the optimization process stable when scaling to larger state and action space, we base our implementation to be similar to the TD3 [17] algorithm. In Appendix C, we outline the learning procedure to learn using the policy gradient derived in Proposition 2. Furthermore, we also discuss how the proposed algorithm handles continuous state spaces.

## C   Neural Network Architecture and Implementation

An implementation of the $\eta\psi$-Learning algorithm together with instructions for reproducing the experiments presented in this paper can be found at `https://github.com/arnavkj1995/Eta_Psi_Learning`.

$\eta\psi$-Learning approximates the SR with a parameterized function $\boldsymbol{\psi_\theta}$ to learn an exploration policy and predict the state visitation distribution. Because the SR is conditioned on a trajectory $\tau$ of variable length, we implement the function $\boldsymbol{\psi_\theta}$ with a Recurrent Neural Network (RNN) architecture, as outlined in Figure 6. In this architecture, the states in a trajectory $\tau_{:T}$ are first fed through a encoder network (E) comprising of Multi-Layer Perceptron (MLP) layers. Subsequently, the output of the Multi-Layer Perceptron (MLP) is fed through an Recurrent Neural Network (RNN) (denoted with F) architecture to compress the state sequence into one real-valued feature vector. Since, RNN are known to suffer from vanishing gradients [7], we implement the RNN with a Gated Recurrent Unit (GRU) [10]. Leveraging recurrent networks to learn the SR has been explored previously in [4, 8]. Finally, the recurrent state obtained from the RNN is concatenated with the representation of the current state and is passed through the the decoder (D) with MLP layers to predict an SR vector for a given action. In the following paragraphs, we elaborate on how the proposed architecture was used to train the agent for finite and infinite action spaces.

**Finite Action Space Variant**    For the finite action space variant, the decoder outputs a SR vector for each action $a \in \mathcal{A}$. This is similar to the prior method that learns Successor Features (SF) for discrete action spaces [33, 3]. Algorithm 2 describes the learning procedure for training $\eta\psi$-Learning to get exploratory policies.

**Infinite Action Space Variant**    For infinite action space variant, the hidden state from the recurrent network is passed through a deterministic actor network which comprises of MLP layers. The policy network (actor) is conditioned on the hidden states because in $\eta\psi$-Learning the policy is a function of trajectories and not individual states. The hidden state from the recurrent network is concatenated with the action to predict the SR vector. The estimated SR vector is used to calculate the visitation distribution over one-hot embeddings of states and these SR predictions are then used to computed to loss objective for optimization. For the Reacher and Pusher tasks, we manually sub-select which dimensions of the state space are one-hot encoded. In these cases, $\eta\psi$-Learning learns an exploration policy that maximizes the entropy of visitation distribution across these sub-selected dimensions only. This approach to sub-selecting state dimensions is similar to prior work on maximum state entropy exploration [23, 43]. In this work, the agent is trained using similar techniques as the TD3 agent [17]. The agent keeps a single encoder and recurrent network to encode the history of

**Algorithm 3** $\eta\psi$-Learning: Infinite Action Space Framework

1: Initialize SR network with parameters $\theta_1, \theta_2$, policy parameters $\mu$ and the replay buffer $\mathcal{B} = \{\}$
2: Set target parameters equal to the main parameters: $\theta_{targ,1} = \theta_1, \theta_{targ,2} = \theta_2$, and $\mu_{targ} \leftarrow \mu$
3: Denote the predecessor feature with $\boldsymbol{\eta}$, discount function with $\gamma$, and episode length with $h$
4: **while** Training **do**
5:    Collect $\tau_{exp} = \{s_1, a_1, .., s_h\}$ using target policy $\pi_{\mu_{targ}}$ and add to replay buffer $\mathcal{B}=\mathcal{B} \cup \tau_{exp}$
6:    **for** each training step $j$ **do**
7:       Sample batch of $\tau = (s_1, ..a_{l-1}, s_l) \sim \mathcal{B}$ of sequence length $l \in \{2, .., h\}$
8:       Compute target actions $a' = clip(\pi_{\mu_{targ}}(\tau_{:l}) + clip(\epsilon, -c, c), a_{Low}, a_{High}), \epsilon \sim \mathcal{N}(0,1)$
9:       Compute i=$\arg\min_{k\in\{1,2\}} H(\boldsymbol{\eta}(\tau_{:l-1}) + \boldsymbol{\psi}_{\theta_k}(\tau_{:l-1}, a'))$
10:      Compute target $\boldsymbol{y}=\boldsymbol{e}_{s_l} + \gamma(l)\,\boldsymbol{\psi}_{\theta_i}(\tau_{:l-1}, a')$
11:      Update the SR networks by performing gradient steps on
$$||\boldsymbol{y} - \boldsymbol{\psi}_{\theta_i}(\tau_{:l-1}, a_{l-1})||_2^2, \qquad i = 1, 2$$
12:      **if** j % policy_update == 0 **then**
13:         Perform update step for policy by computing gradients using
$$\textstyle\sum_i z_i \nabla_a \boldsymbol{\psi}_{\theta_1}(\tau_{:l}, a) \,|_{a=\pi(\tau_{:l})} \nabla_\mu \pi_\mu(\tau_{:l}),$$
where $z_i = -\log(\boldsymbol{\eta}(\tau_{:l})_i - \boldsymbol{\psi}_{\theta_1}(\tau_{:l}, \pi_\mu(\tau_{:l}))_i) + 1$
14:         Update target networks with
$$\theta_{targ,i} \leftarrow \rho\theta_{targ,i} + (1 - \rho)\theta_i, \qquad i = 1, 2$$
$$\mu_{targ} \leftarrow \rho\mu_{targ} + (1 - \rho)\mu$$
15:      **end if**
16:   **end for**
17: **end while**

observed states. The encoded states are passed through two decoder networks to predict the SR vectors, which are used to represent the two critic networks. The target for SR is computed using the vector that leads to a smaller value of the two utility functions. There is a single actor network that specifies the action from the hidden state. In addition, $\eta\psi$-Learning-maintains a target network for all the components—encoder, recurrent, critics, and actor networks. Furthermore, similar to the TD3 algorithm, a clipped noise is added to each dimension of the action from the target network. Moreover, we also use delayed actor updates where the actor network is updated less frequently than the SR networks. Lastly, the gradients from the actor are not passed through the encoder and the recurrent networks. The procedure for training this variant is provided in Algorithm 3.

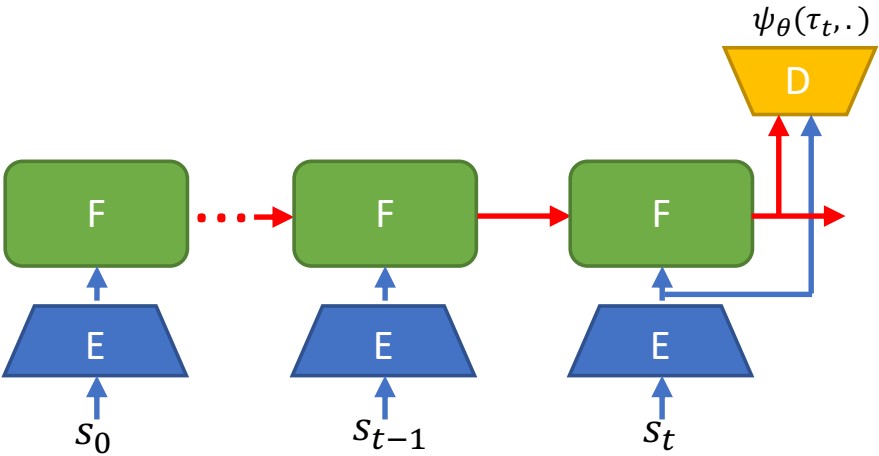

Figure 6: Network architecture to learn the SR. The states are firstly passed through an encoder (E), followed by feeding the encoded states through a RNN (GRU in our case) (F). This compresses the history of visited states, and the obtained hidden state is concatenated with the encoded state to predict the SR vector for an action using a decoder network (D).

# D   Limitations

In this work, we developed an algorithm to learn exploratory policies at convergence that can explore the state space efficiently within a finite-length trajectory. Such policies can benefit generalization in different applications like Meta-RL and episodic exploration. Maximum state entropy exploration is a potential direction for learning such policies. However, prior works are not very efficient as they either learn a Markovian policy, optimize for the state coverage over multiple long trajectories, or learn a mixture of stochastic policies. Due to these shortcomings, they are not widely used for solving tasks in RL. To address these concerns, we introduce $\eta\psi$-Learningand demonstrate that the proposed algorithm can learn to efficiently explore the state space within a finite length trajectory. $\eta\psi$-Learning achieves this by combining predecessor and successor representation to estimate the state-visitation distribution and utilizing this to optimize the entropy-based objective. [43] theoretically showed that learning such policies that achieve zero-regret is NP-Hard and we develop a practical algorithm to solve such tasks. We hope that the proposed method bridges the gap of leveraging policies learned using maximum state entropy exploration for more complex tasks in RL. As with any new approach, there are certain limitations:

- *Scaling to high-dimensional inputs*: Learning to explore more complex tasks with high-dimensional input spaces would require using a better representation learning method and a mechanism to estimate successor and predecessor representations. For representation learning, existing methods that use auxiliary losses, inverse/forward dynamics, or random network-based features can be used. The more challenging task is learning the SR and future works can explore leveraging methods like Successor Measures [62] or ProtoValueNetworks [15].

- *Environments with changing dynamics*: The learned SR depends on the environment dynamics and the policy, and we learn SR for a fixed environment in this work. However, many real-world tasks require exploration in an environment with changing dynamics (procedural environments [24, 67]). A potential direction is learning universal successor representation approximators [8] where the successor representations are conditioned on a context that defines the environment and we leave this for future research.

- *Architectural priors for estimating SRs*: The successor representations use an RNN which is known to suffer from vanishing gradient problems. Many real-world tasks require agents to retain information over multiple timesteps. Future research can explore having better architectural priors like Transformers or S4 that have better memory and are known to work well on complex tasks.

- *Estimating predecessor representation*: In this work, we computed the predecessor representation as the summation of the prior state representations. However, recent methods like Expected Eligibility Traces [64] show better sample efficiency and we leave leveraging such methods for future research.

- *Computing entropy instead of state visitation distribution*: While predicting future state visitations may seem harder than needed for entropy prediction, it is important to note that optimal decision depends on which states are visited multiple time steps into the future. It is possible that there exist more efficient algorithms for predicting this entropy, but to our knowledge such algorithms do not appear in the published literature, and $\eta\psi$-Learning is the first of this kind. Another challenge with estimating entropy is that the estimator needs to adapt to the changing policy during training. In the proposed algorithm, this is mitigated as both the entropy and policy directly depend on the estimated SR vector requiring no additional updates to estimate entropy given the policy. However, future work would involve discovering other more efficient methods for estimating the entropy induced by the future state visitations.

- *Comparison to Forward-Backward Representation [63]*: The Forward-Backward representations [62, 63] capture similar state visitation statistics as the predecessor and successor representation used by $\eta\psi$-Learning. Mathematically, the forward-backward (FB) representation factorizes the Q-function in an RL setting in a very different way than the $\eta\psi$-Learning algorithm. The FB method also focuses on a reward-maximization setting instead of exploration and are only conditioned on states whereas the $\eta\psi$-Learning algorithm conditions these representations on trajectories. During training, the focus of FB learning is on representation learning using an offline dataset where it is assumed that the agent does

not have to explore the environment. The learned FB representations are then used to solve multiple different tasks using the same representations, assuming that reward parameterization is known. However, the $\eta\psi$-Learning algorithm learns exploratory policies that could be used to initially explore an unknown task to determine this reward parameterization. We believe that the $\eta\psi$-Learning algorithm complements the FB method and integrating the two systems into a cohesive RL agent is an interesting avenue for future research.

## E   Discount Function

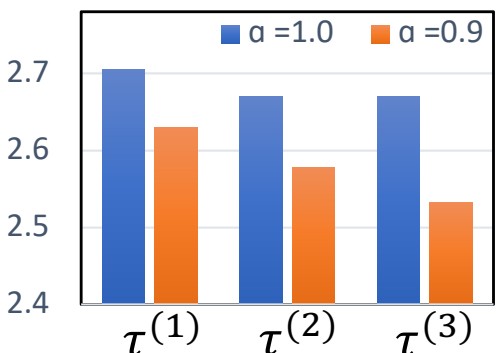

Figure 7: Illustration of the values of the entropy with different value of $\alpha$ hyperparameter in the proposed $\gamma$-function for the trajectories introduced in Figure 1.

In this work, we have used a time-dependent $\gamma$-function. Using the gridworld example described in Figure 1, we now present how the choice of $\gamma$-function affects the entropy term in the objective. Suppose there are three trajectories followed by the given trace $\tau_{:T}$, where we denote the $i$-th trajectory with $\tau^{(i)}$. Here, Figure 1(b) shows an optimal trajectory ($\tau^{(1)}$) which combined with the trace covers each cell of the grid with 15 steps. Figure 1(c) presents a suboptimal trajectory ($\tau^{(2)}$) where the agent takes the right action from the current state and visits a previously observed state in the last step. Figure 1(d) shows another sub-optimal trajectory ($\tau^{(3)}$) which takes the right action in the current state but visits the new state twice because it goes to the top right corner of the grid.

For the intermediate step $T$, we define the discount factor for the predecessor representation for the trace as $\gamma(t) = \frac{\alpha^{T-t}}{Z}$, where $\alpha$ is a scalar between (0, 1], and $Z = \Sigma_{t=0}^{T}\alpha^{T-t} + \Sigma_{t=T}^{h}\alpha^{t-T}$ is the normalization factor. The $\gamma$-function for the successor representation is denoted using $\gamma(t) = \frac{\alpha^{t-T}}{Z}$. The proposed $\gamma$-function for both the representations is similar to discounting used in standard RL literature [13, 59]. Upon comparing the entropy for given trajectories with $\alpha = 0.9, 1.0$, we observe in Figure 7 that $\tau^{(1)}$ being the optimal trajectory attains higher entropy when compared with $\tau^{(2)}$ and $\tau^{(3)}$. The other sub-optimal trajectories $\tau^{(2)}$ and $\tau^{(3)}$ achieve same entropy when $\alpha$ is set to 1.0. However, for $\alpha = 0.9$, the discount function $\gamma$ emphasizes which states are visited earlier in the trajectory and assigns the lowest score to the trajectory $\tau^{(3)}$ because this trajectory revisits states earlier in the sequence than the other options $\tau^{(2)}$ and $\tau^{(1)}$. This example illustrates how the $\gamma$-function can be used to trade off near-term vs. long-term exploration behavior. Depending on the $\alpha$ setting, the agent can be encouraged to avoid re-visiting states either only in the short-term or the long-term, similar to how discounting encourages maximizing short-term over long-term rewards in algorithms like Q-learning.

## F   Environments

For finite action space variant, we experimented with ChainMDP, RiverSwim, $5 \times 5$ Grid-world, TwoRooms and FourRooms environments, which have finite action and state space, respectively. For infinite action space variant, we experiment with Reacher and Pusher environments where we want

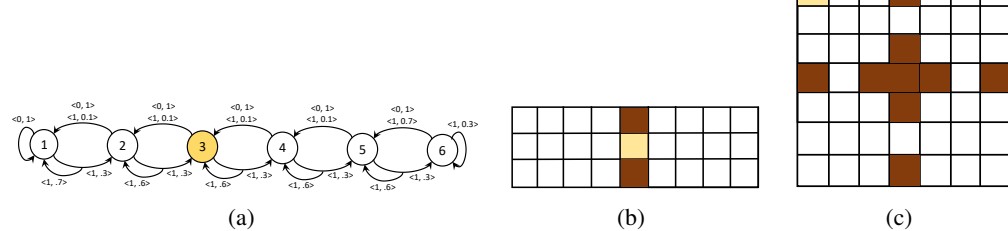

(a)             (b)             (c)

Figure 8: (a) RiverSwim [58], (b) TwoRooms, and (c) FourRooms environments, respectively. The yellow block denote the initial state of the agent in each episode. The brick red regions represent the walls in the environment. When taking an action that collides with the walls, the chosen action does not change the state of the agent.

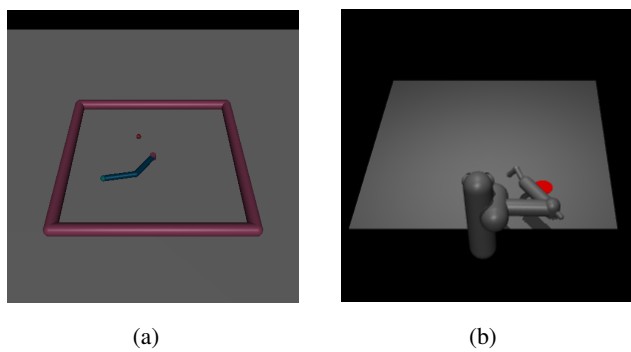

(a)                          (b)

Figure 9: (a) Reacher and (b) Pusher environments for experiments with infinite action space.

the agent to move its fingertip to different locations in the environment. The environments used in this work are further described below:

**ChainMDP:** ChainMDP is an environment where the agent can take only move in two directions— left or right. In this work, we experiment with both deterministic and stochastic version of ChainMDP environments. The stochastic ChainMDP is similar to RiverSwim environment [58] (Figure 8(a)).

**GridWorld:** In the Gridworld environment (shown in Figure 1), the agent can take 4 actions to move in any of the 4 directions. In this work, we experiment with the gridworld of dimensions $5 \times 5$. The agent always start in the top left corner of the grid. For the gridworld, there are multiple possible optimal trajectories, and the number of such trajectories increases explonentially with size of grid.

**TwoRooms:** The proposed TwoRooms environment is a gridworld with some walls. As shown in Figure 8(b), the agent starts at the center of wall between the two rooms and has to first navigate in one of the rooms, visit the starting state and then move to the other room. This makes the task challenging as the agents requires to track the trace because when the agents reaches initial state after exploring one room, the information of which room was visited should aid in going to the other room.

**FourRooms:** The FourRooms environment (depicted in Figure 8(c)) has four rooms connected which are connected by open shots between the walls. This task is even more challenging as the agents while navigating need to first explore the current room followed by efficiently going across different rooms.

**Reacher:** The Reacher environment (shown in Figure 9(a)) is a continuous control environment having a two-jointed robotic arm with continuous state and action spaces. The action space denotes the torques applied to the hinges. The state denotes the position, angles and angular velocities of the arms. The agent is tasked to maximize the entropy over the position of the fingertip.

**Pusher:** The Pusher environment (shown in Figure 9(b)) is a continuous control environment having a multiple-jointed robotic arm with continuous state and action spaces. The action space denotes the torques applied to the hinges. The state denotes the position, angles and angular velocities of the arms/hinges. Similar to the Reacher environment, the agent is tasked to maximize the entropy over the position of the fingertip. However, this task is harder to solve because having multiple joints leads to a larger action and state space making it a more challenging control problem.

For training and evaluation, we have used different parameters specific to each of the environment. For training, the parameters are the length of an episode and the number of episodes used for training. The number of environment steps can be obtained by multiplying these 2 parameters. For evaluation of an agent, we use different episode length for the three metrics defined to measure the performance of agents in section 5.

| Name | ChainMDP | RiverSwim | Gridworld | TwoRooms | FourRooms |
|---|---|---|---|---|---|
| *Training Parameters* | | | | | |
| Length of trajectory from environment | 20 | 50 | 50 | 100 | 200 |
| Number of episodes | 1000 | 1000 | 1000 | 1000 | 2500 |
| *Evaluation Parameters* | | | | | |
| Horizon $h$ for Entropy metric | 20 | 50 | 50 | 100 | 200 |
| Horizon $h$ for State Coverage metric | 20 | 50 | 50 | 100 | 200 |
| Horizon $h$ to measure Episode Length | 100 | 500 | 200 | 1000 | 1000 |

Table 1: Defines the parameters of the environments with discrete actions during training and evaluation, respectively.

| Name | Reacher | Pusher |
|---|---|---|
| *Training Parameters* | | |
| Length of trajectory from environment | 100 | 200 |
| Number of episodes | 1000 | 1000 |
| *Evaluation Parameters* | | |
| Horizon $h$ for Entropy metric | 100 | 200 |
| Horizon $h$ for State Coverage metric | 100 | 200 |

Table 2: Defines the parameters of the environments with continuous actions during training and evaluation, respectively.

# G   Hyper Parameters

In this section, we describe the hyperparameters used for training the proposed method $\eta\psi$-Learning. Table 3 and Table 4 presents the list of hyperparameters for the discrete and continuous control environments, respectively. All models were trained on a single NVIDIA V100 GPU with 32 GB memory. The implementation of the proposed method was done using the RLHive [46] library.

# H   Meta-RL

In this work, we demonstrated how $\eta\psi$-Learning can learn optimal policies that can maximize the entropy of state visitation distribution. Such policies are useful for many subareas of RL where during evaluation the task is to adapt to new reward functions with minimal interactions with the environment. A challenging subproblem in such tasks is to infer the reward function. This is especially harder when the reward is sparse. Some recent works have explored adding exploratory behaviors for initial interactions with the environment to allow agent to infer the reward function. VariBAD [70] algorithm learns optimal policies that can explore during evaluation to speed up adaptation for Meta-RL tasks. VariBAD maintains a belief over the state space to explore in the environment and upon discovering the reward function adapts to the To illustrate the exploratory capabilities of $\eta\psi$-Learning, we compare with VariBAD as baseline in this section.

| Name | Value |
|---|---|
| Batch Size | 32 |
| Sequence Length | 10 / 20 / 50 / 50 / 100 |
| $\alpha$ for $\gamma$-function | 0.95 |
| Encoder layers | 1 |
| Encoder output dimensions | 64 / 64 / 128 / 128 / 256 |
| Encoder activation | LeakyReLU [68] |
| Hidden state of GRU | 64 / 64 / 128 / 128 / 256 |
| Hidden layer dimension for SR decoder | 32 / 32 / 64 / 64 / 128 |
| Decoder activation | None |
| Optimizer | Adam [29] |
| Learning rate | 3e-4 |
| Capacity of replay buffer | 200000 |

Table 3: Hyper parameters used for training $\eta\psi$-Learning. When parameters are separated by /././././., it means the corresponding hyperparameters for ChainMDP, RiverSwim, Gridworld, TwoRooms and FourRooms environments, respectively. When tuning the agent for a task, we recommend searching over $\alpha \in \{0.9, 0.95, 0.98, 0.99\}$, and hidden state dimension of GRU and encoder output dimensions in $\{64, 128, 256, 512\}$. For the replay buffer, we have used the replay buffer implemented in DreamerV2 [20], which for an episode samples a chunk of a given length.

| Name | Value |
|---|---|
| Batch Size | 256 |
| Sequence Length | 100 |
| $\alpha$ for $\gamma$-function | 0.95 |
| Encoder layers | 2 |
| Encoder output dimensions | 256 |
| Encoder activation | LeakyReLU [68] |
| Hidden state of GRU | 256 |
| Hidden layer dimension for SR decoder | 256 |
| Decoder activation | None |
| Optimizer | Adam [29] |
| Learning rate | 3e-4 |
| Capacity of replay buffer | 200000 |
| Polyak constant | 0.005 |
| Grad Clip | 5.0 |
| Action noise | 0.1 |
| Target noise | 0.2 |

Table 4: Hyper parameters used for training $\eta\psi$-Learning on continuous state space environments. When tuning the agent for a task, we recommend searching over $\alpha \in \{0.9, 0.95, 0.98, 0.99\}$, and hidden state dimension of GRU and encoder output dimensions in $\{64, 128, 256, 512\}$. For the replay buffer, we have used the replay buffer implemented in DreamerV2 [20], which for an episode samples a chunk of a given length.

The implementation of VariBAD provided by the authors is used to conduct this experiment. The baseline was trained for 10 million environment steps on the $5 \times 5$ gridworld using the setup described in the paper. Two metrics are employed to compare the agents:

- *State Coverage* computes the fraction of the state space covered by the agent.

- *Goal Search Time* computes the environment steps taken to locate the sparse reward goal state. This evaluates the ability of the agent at quickly finding the reward which is essential for swift adaptation to novel tasks.

The two metrics evaluate the agents on average time taken to find the reward function and the average search completion time for covering the grid. The VariBAD [70] algorithm considered sparse reward task where a random location is sampled after each episode as the goal state and is assigned a high

reward. To compute the *Goal Search Time* metric, we sample a goal state randomly and record the steps taken to locate the target state. For evaluation, the metric is averaged over 16 sampled goal state for each seed. Figure 4(c) presents the comparison of VariBAD and $\eta\psi$-Learning on both metrics across 5 seeds. The proposed method $\eta\psi$-Learning achieves outperforms VariBAD on both metrics while only being trained for 100K environment steps. This demonstrate the efficacy of $\eta\psi$-Learning at exploring in environment that involves inferring the reward function during evaluation. We believe $\eta\psi$-Learning can be combined with a adaptation policy, where the proposed method can explore to find the reward and the adaptation policy is trained to adapt quickly to the reward function, and we leave this for future work.

# I  Ablation Studies

In this section, we present ablations studies to understand the gains of the proposed method $\eta\psi$-Learning.

## I.1  MaxEnt with a recurrent network

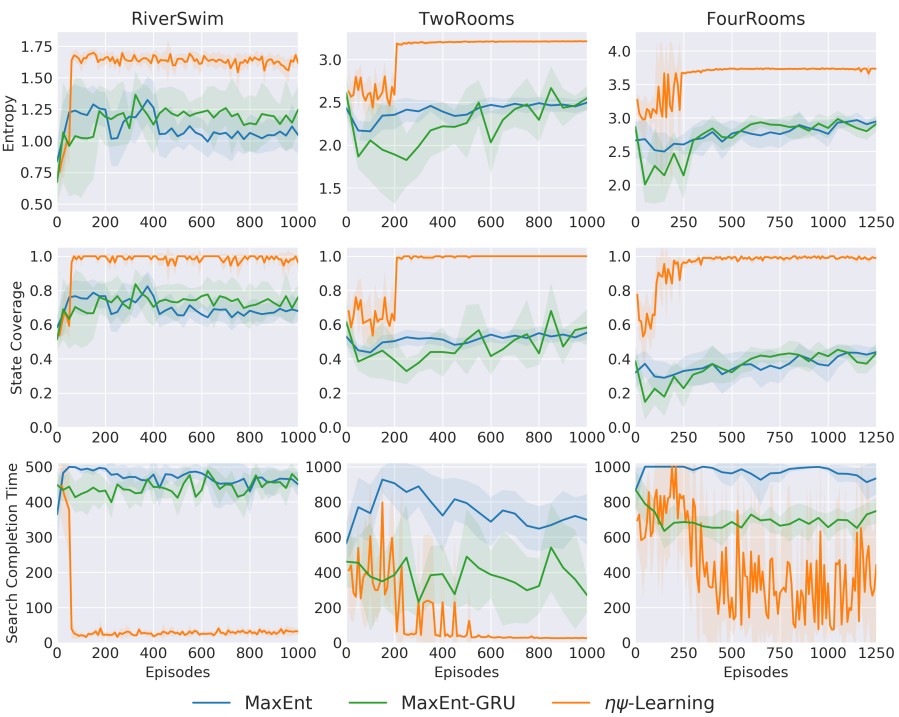

Figure 10: Comparison of the baseline MaxEnt when trained with a recurrent network.

We conduct an experiment with a modification to the baseline MaxEnt [23] where agent observed the history of visited states. This is done to evaluate if the improvements are coming from having a recurrent policy. To this end, the state-conditioned policy in MaxEnt is replaced with a recurrent policy where the GRU [10] encodes the states observed in the trajectory. The parameters of the recurrent policy is optimized using the loss function described in MaxEnt [23]. Figure 10 presents the results of MaxEnt with a recurrent policy (named MaxEnt-GRU) where no gains are observed by having a recurrent policy and the proposed objective function used to train $\eta\psi$-Learning is crucial for learning optimal behaviors.

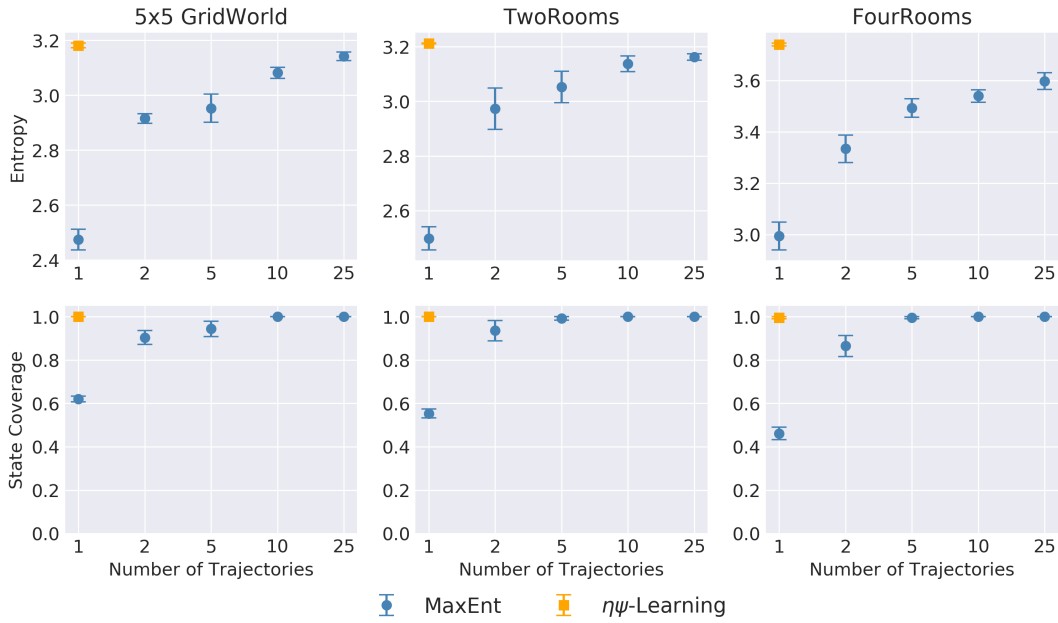

Figure 11: Comparison of $\eta\psi$-Learning with baseline MaxEnt when metrics are computed using multiple trajectories. MaxEnt-X denotes the evaluation with X trajectories.

## I.2 Comparison across multiple trajectories

Contemporary methods on Maximum State Entropy Exploration [23, 42] were evaluated by averaging the state visitation distribution over multiple trajectories. In this work, we demonstrate that $\eta\psi$-Learning can achieve optimal behaviors over a single trajectory of finite length. In this study, we also explore comparison of the baseline MaxEnt when evaluated over multiple trajectories. For this evaluation, we sample a batch of trajectories and then average the state visitation distribution of trajectories. The metrics are then computed using this averaged visited state distribution. In Figure 11, we compare the Entropy and State Coverage over this averaged distribution. MaxEnt-X denotes the metric of MaxEnt after sampling X trajectories during evaluation. The proposed method $\eta\psi$-Learningwas evaluated using a single trajectory. We do not report metrics of $\eta\psi$-Learningacross multiple trajectories as we observed that the gains do not vanish with an increasing number of trajectories. The metrics for the MaxEnt algorithm improve with the increasing number of trajectories used for evaluation. With 10 or more trajectories, the baseline achieves optimal State Coverage. However, $\eta\psi$-Learning achieves full coverage with a single trajectory demonstrating the efficiency of the exploration policies learned using the proposed method. The baseline MaxEnt show similar behaviors by improving on the Entropy metric with more trajectories used for evaluation, whereas $\eta\psi$-Learning still outperforms the baseline when evaluated using a single trajectory. This demonstrates that the proposed method explores the state-space with near-equal state visitations to maximize the entropy while having optimal state coverage in a single trajectory.

## I.3 Effect of the $\alpha$ parameter

We also study the effect of the hyper-parameter $\alpha$ of the $\gamma$-function (discussed in Appendix E). We note that $\alpha$ can be selected using the same method used to select the discount factor in the standard RL. We conducted experiments with $\alpha$={0.8,0.9,0.95,0.99} across three environments- RiverSwim, TwoRooms, and FourRooms (Figure 12). On the RiverSwim environment, all methods converged with similar values across all metrics. On the TwoRooms environments, agents with $\alpha$={0.8, 0.99} were not performing well across the three metrics. Moreover, the convergence was slower for agent with $\alpha = 0.9$ when compared with agent trained with $\alpha = 0.95$. Our intuition behind this is that when $\alpha$ is smaller, the memory of the visited states in predecessor representation ($\eta$) reduces leading to a re-visitation of observed states. Whereas when $\alpha$ is large, then the agent does similarly discount a future state at any point in the trajectory. Lastly, on the FourRooms environment, the results are similar to the TwoRooms environment but with more pronounced differences in metrics for different

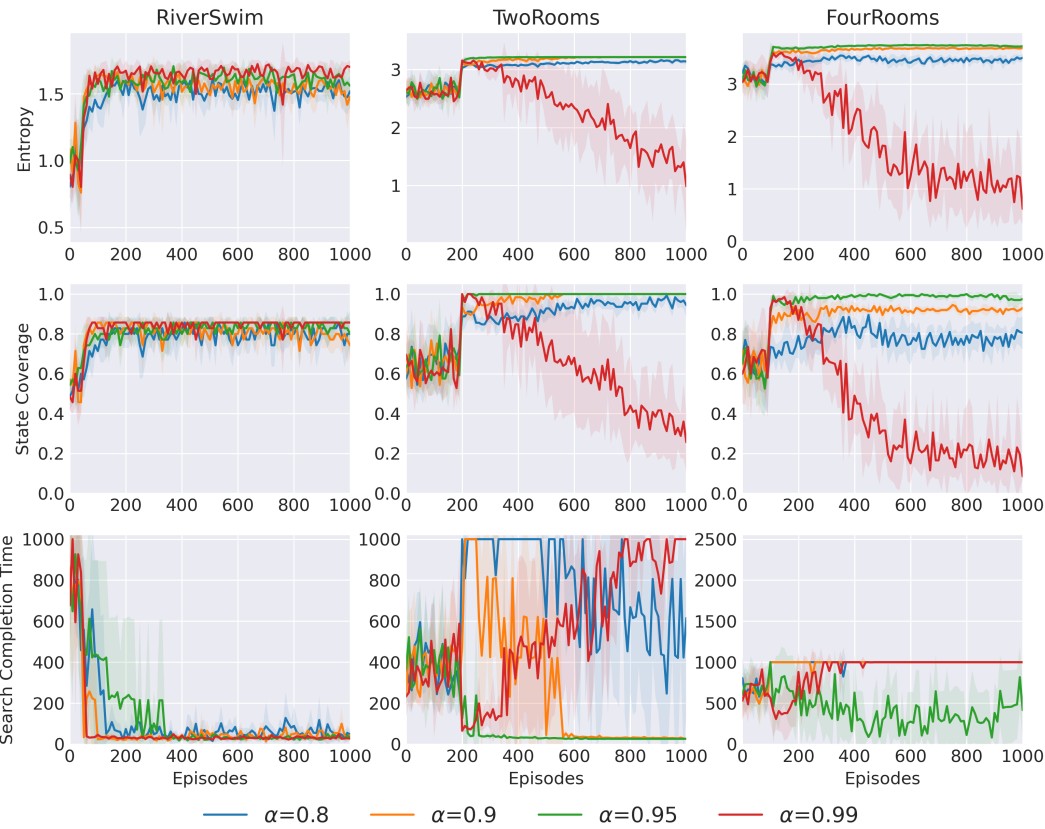

Figure 12: Evaluation with different value of $\alpha$ hyperparameter in the proposed $\gamma$-function.

values of $\alpha$. This is because FourRooms environment is harder to solve than TwoRooms environments. Notably, the Search Completion Time metric diverges for all values of $\alpha \neq 0.95$, which shows that $\alpha = 0.95$ leads to optimal behavior for our task. For new tasks, we believe $\alpha$ can be tuned similarly to the discount factor used in standard RL.

### I.4 SR conditioned on state

In this section, we validate if conditioning on the history of visited states is important for learning optimal SR vectors. We consider a variant of $\eta\psi$-Learning where the SR is only conditioned on the current state. However, the objective proposed in Equation 7 which uses predecessor representation to compute the entropy is used for learning. We conduct experiments on continuous control tasks where we observe that conditioning on the entire trajectory sequence is important for learning optimal SR vectors (Figure 13).

### I.5 Visualization of trajectories

The maneuvers taken by the learned $\eta\psi$-Learning agent were also visualised. For the Reacher environment, the agent was first covering the faraway states followed by covering the nearby states to the central position. Similar behaviors were observed for the Pusher environment, where the agent was moving the fingertip to different locations on the table top and tried visiting all reachable locations on the table.

## J Comparison on Sparse Mountain Car

In this work, we showed how the proposed method $\eta\psi$-Learning can learn policies to efficiently explore in an environment. To present the applicability of the proposed method to other settings,

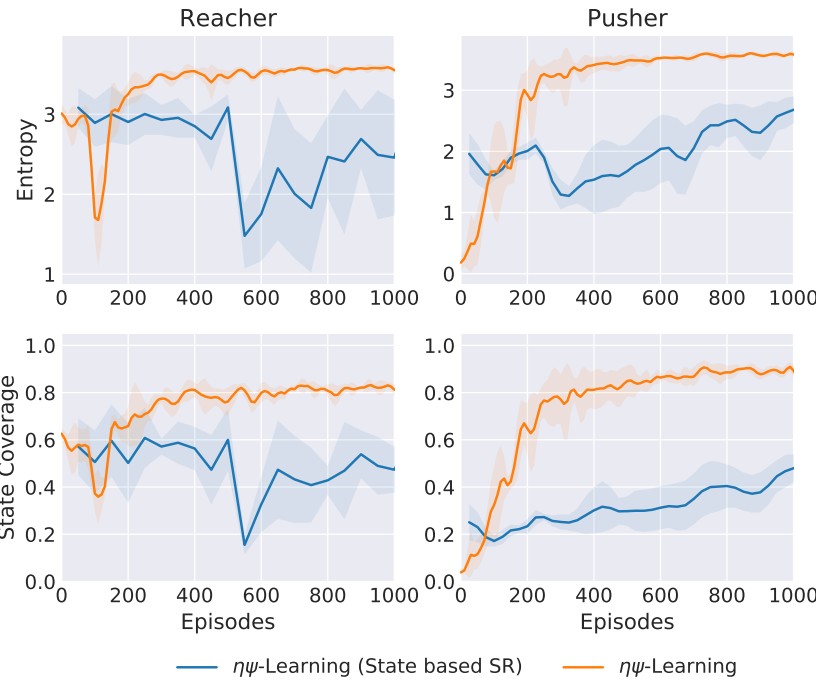

Figure 13: Comparison with a proposed variant where the SR is conditioned on the state only denoted as $\eta\psi$-Learning (State based SR). We see that conditioning on the entire trajectory is important for learning SR vectors to optimize the objective.

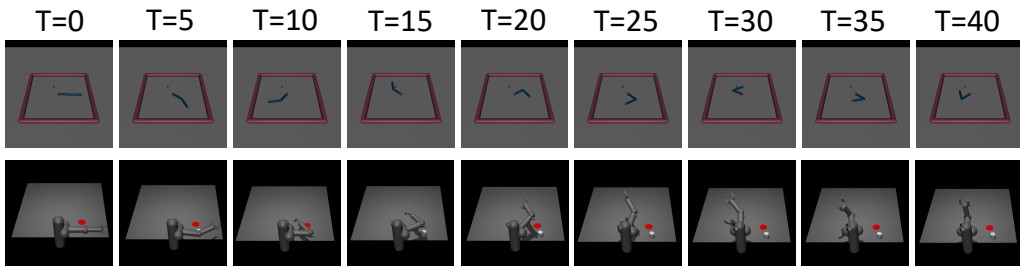

Figure 14: Rolled out trajectories using the learned $\eta\psi$-Learning agent on the Reacher (top row) and Pusher (bottom row) environments at different time steps, respectively.

we conduct an experiment on tasks with sparse rewards. We consider the Sparse MountainCar environment where the agent receives a positive reward after reaching the goal position after which the episode terminates and the agent receives no reward in other states. We chose this environment because the reward is sparse, and it is hard for the agent to discover the reward function as the agent needs to plan to explore the top of the hill. For the baselines, we compare with a random agent, TD3 [17], TD3 combined with count-based bonus [6] as an intrinsic reward (TD3-Count), and TD3 combined with first occupancy bonus [40] as an intrinsic reward (TD3-First).

We also propose a variant of our method to learn with extrinsic rewards. We modified the TD3 algorithm and call the proposed variant TD3-$\eta\psi$-Learning. In this variant, we propose to learn 2 critics- one to estimate the SR as mentioned in 4 ($Q^{expl}$), and the other to estimate the returns obtained using extrinsic rewards ($Q^{ext}$). The critic based on extrinsic rewards $Q^{ext}$ is conditioned on the current state learned using the update rule defined in TD3 [17]. Notably, $Q^{expl}$ and $Q^{ext}$ are conditioned on the trajectory of prior visited states and current state, respectively. To learn the actor, the gradients are obtained using the overall Q-function which is defined as the linear sum of both Q-values based on extrinsic rewards and entropy-based term: $Q = Q^{ext} + \beta Q^{expl}$, where $\beta$ denotes

---

**Algorithm 4** $\eta\psi$-Learning: TD3-$\eta\psi$-Learning

---

1: Initialize SR network with parameters $\theta_1$, $\theta_2$, $Q^{ext}$ network with parameters for Q-function with extrinsic rewards $\phi_1$, $\phi_2$, policy parameters $\mu$ and the replay buffer $\mathcal{B} = \{\}$
2: Set target parameters equal to the main parameters: $\theta_{targ,1} = \theta_1$, $\theta_{targ,2} = \theta_2$, $\phi_{targ,1} = \phi_1$, $\phi_{targ,2} = \phi_2$, and $\mu_{targ} \leftarrow \mu$
3: Denote the predecessor feature with $\boldsymbol{\eta}$, discount function with $\gamma$, and episode length with $h$
4: **while** Training **do**
5:     Collect $\tau_{exp} = \{s_1, a_1, r_1, .., s_h\}$ using $\pi_{\mu_{targ}}$ and add it to replay buffer $\mathcal{B} = \mathcal{B} \cup \tau_{exp}$
6:     **for** each training step $j$ **do**
7:         Sample batch of $\tau = (s_1, .., a_{l-1}, r_{l-1}, s_l) \sim \mathcal{B}$ of sequence length $l \in \{2, .., h\}$
8:         Compute target actions $a' = clip(\pi_{\mu_{targ}}(\tau_{:l}) + clip(\epsilon, -c, c), a_{Low}, a_{High}), \epsilon \sim \mathcal{N}(0, 1)$
9:         Compute i=$\arg\min_{k\in\{1,2\}} H(\boldsymbol{\eta}(\tau_{:l-1}) + \boldsymbol{\psi}_{\theta_k}(\tau_{:l-1}, a'))$
10:        Compute target for SR
$$\boldsymbol{y}_\theta = \boldsymbol{e}_{s_l} + \gamma(l)\, \boldsymbol{\psi}_{\theta_{targ,i}}(\tau_{:l-1}, a')$$
11:        Update the SR networks by performing gradient steps on
$$\|\boldsymbol{y}_\theta - \boldsymbol{\psi}_{\theta_i}(\tau_{:l-1}, a_{l-1})\|_2^2, \qquad i = 1, 2$$
12:        Compute target for $Q^{ext}$
$$\boldsymbol{y}_\phi = r_{l-1} + \gamma_{ext} \min_{k\in 1,2} Q^{ext}_{\phi_{targ,k}}(s_l, a')$$
13:        Update the $Q^{ext}$ networks by performing gradient steps on
$$\|\boldsymbol{y}_\theta - Q^{ext}_{\phi_i}(s_{l-1}, a_{l-1})\|_2^2, \qquad i = 1, 2$$
14:        **if** j % policy_update == 0 **then**
15:            Perform update step for policy by computing gradients using
$$\sum_i z_i \, \nabla_a \boldsymbol{\psi}_{\theta_1}(\tau_{:l}, a) \,|_{a=\pi(\tau_{:l})} \, \nabla_\mu \, \pi_\mu(\tau_{:l}) + \nabla_\mu Q_{\phi_1}(s_{l-1}, \pi_\mu(\tau_{l-1})),$$
           where $z_i = -\log(\boldsymbol{\eta}(\tau_{:l})_i - \boldsymbol{\psi}_{\theta_1}(\tau_{:l}, \pi_\mu(\tau_{:l}))_i) + 1$
16:            Update target networks with
$$\theta_{targ,i} \leftarrow \rho\theta_{targ,i} + (1 - \rho)\theta_i, \qquad i = 1, 2$$
$$\mu_{targ} \leftarrow \rho\mu_{targ} + (1 - \rho)\mu$$
17:        **end if**
18:     **end for**
19: **end while**

---

the trade-off between the two $Q$-functions. We have added a pseudo-code to describe the proposed algorithm in Algorithm 4.

Figure 5 presents the results on the Sparse Mountain Car environment. We compare two metrics-Return and Episode Length denoting the steps taken to reach the goal state across 5 seeds. The Average Steps metric highlights if the agent learns to solve the task with minimal interactions. We plot the mean and 95% confidence interval in shading. The proposed method outperforms TD3 and variants with count-based and first occupancy based bonuses. Through our experiment on Sparse MountainCar, we demonstrate that the proposed method can improve efficiency in standard RL tasks, especially in sparse reward environments. We leave it for future work to leverage the proposed extension in POMDP setting with more complex input spaces.

