# OpenReview forum: "Maximum State Entropy Exploration using Predecessor and Successor Representations"
_NeurIPS.cc/2023/Conference — NeurIPS 2023 poster_

### Official Review · Reviewer_RqRN · 2023-06-26

**Soundness:** 3 good
**Presentation:** 4 excellent
**Contribution:** 2 fair
**Rating:** 6
**Confidence:** 4

**Summary:**

This paper addresses the problem of maximum state entropy exploration in environments without rewards. Especially, it proposes a method to learn an history-based policy maximizing the entropy over the states sampled in a single trajectory. The method, called $\eta\psi$-Learning, combines predecessor representations, to keep memory of the previously visited states, and successor representations, to predict the states that will be visited in expectation under the current policy. The introduced algorithm is tested against MaxEnt in a set of toy domains and some continuous control tasks, it is also briefly tested against VariBAD in gridwolrd domain.


**Strengths:**

- The paper proposes a simple yet effective method to learn non-Markovian policies that maximize the state entropy induced by a single trajectory;
- The introduced method remarkably improves the performance of MaxEnt, here taken as a representative algorithm for maximum state entropy exploration with Markovian policies;
- The ideas are presented with clarity and with some nice visualizations (e.g., Fig. 1 and 3) to support the intuition.


**Weaknesses:**

- It is unclear how the procedure can be scaled up to more challenging domains, e.g., image-based inputs;
- The method is only tested against MaxEnt, which is not the state-of-the-art method for state entropy maximization, especially in continuous control tasks.

**Questions:**

This is an overall interesting paper, tackling a relevant problem raised by (Mutti et al., 2022a) on how to learn non-Markovian policies for maximum state entropy exploration efficiently. Especially, the authors propose to circumvent the inherent computational intractability of the problem with a well-motivated function approximation approach, in which the current decision is conditioned on a compact representation of the history and a forecast of the future state visitations under the policy.
Although the paper does not introduce particularly novel or surprising ideas, it presents the method with clarity and showcases a brilliant performance for $\eta\psi$-Learning, at least in comparison with MaxEnt. While those might be sufficient reasons for acceptance, I believe that including a discussion on how this procedure can be scaled up to more challenging domains, such as image-based tasks, would make for a significant leap in the value of the paper.

*(Major. How to scale up the approach?)*
The experiments show a remarkable improvement over MaxEnt, but previous works (e.g., Mutti et al., 2021) have demonstrated how Markovian policies maximizing the state entropy can be learned in challenging domains. Instead, it is unclear how the predecessor and successor representations of $\eta\psi$-Learning can be scaled to truly high-dimensional settings. While the method may be trivially adapted to continuous settings via discretization, it is unclear whether it can match the performance of previous works based on $k$-NN entropy estimators. Can the authors discuss avenues to scale their approach to high-dimensional domains? Estimating $\psi_\theta$ in practice would be significantly easier than state density estimation?

*(Estimating the state visitations instead of the entropy)*
In $\eta\psi$-Learning, the quality of an action is evaluated in terms of the future visitations it will induce under the policy, from which the entropy of the trajectory can be then computed. However, the task of estimating the future state visitations seems strictly harder w.r.t. what is really needed to make an optimal decision, i.e., an estimate of the entropy that future visitations will induce. I am wondering whether there is a way to directly target entropy estimation instead of future state visitations in $\eta\psi$-Learning.

*(Experiments. MaxEnt performance)*
The learning curve of MaxEnt is quite underwhelming: While the inferior performance of Markovian policies against $\eta\psi$-Learning is reasonable, I would have expected to see some learning progress for MaxEnt as well, especially in domains in which the random policy is far from the optimal strategy, such as in ChainMDP and RiverSwim. Can the authors clarify what is happening with the learning curve of MaxEnt?

*(How the approach relates to Forward-Backward representations?)*
A previous paper (Touati and Ollivier, Learning one representation to optimize all rewards, 2021) also addressed unsupervised reinforcement learning through representations of the past and future visitations, which they call forward-backward representations, bearing some similarity with the predecessor and successor representations of $\eta\psi$-Learning. Differently from this paper, they make learning these representations the actual unsupervised objective, and they advocate for directly employing forward-backward representations for zero-shot adaptation to the supervised task, instead of running state entropy maximizing policies first. Can the author compare $\eta\psi$-Learning with this previous approach, and explain whether the successor representation might be directly exploited in a supervised adaptation phase?

**Limitations:**

The paper does not include an explicit assessment of the limitations of the presented approach, which would add significant value to the paper in my opinion.

---

> ### Author Rebuttal · Authors · 2023-08-09
>
> We thank the reviewer for their time and valuable feedback. We aim to address the concerns in the following:
> > MEPOL as baseline
>
> We thank the reviewer for pointing this out and have added MEPOL[1] as baseline (Figure 1 of rebuttal PDF) and observe that $\eta\psi$-Learning outperforms MEPOL on continuous control tasks of Reacher and Pusher when compared over a single trajectory of finite length.
> > Major. How to scale up the approach?
>
> Learning to explore more complex tasks with high-dimensional input spaces would require using a better representation learning method and a mechanism to estimate successor and predecessor representations. For representation learning, existing methods that use auxiliary losses, inverse/forward dynamics, or random network-based features can be used. The more challenging task is learning the SR and future works can explore using Successor Measures[2] and ProtoValueNetworks[3].
>
> > Estimating the state visitations instead of the entropy
>
> Thank you for raising this point. While predicting future state visitations may seem harder than needed for entropy prediction, it is important to note that optimal decision depends on which states are visited multiple time steps into the future. It is possible that there exist more efficient algorithms for predicting this entropy, but to our knowledge such algorithms do not appear in the published literature and $\eta\psi$-Learning is the first of this kind. Another challenge with estimating entropy is that the estimator needs to adapt to the changing policy during training. In the proposed algorithm, this is mitigated as both the entropy and policy directly depend on the estimated SR vector requiring no additional updates to estimate entropy given the policy. In the revised version, we have highlighted this aspect and state that future work would involve discovering other more efficient methods for estimating the entropy induced by future state visitations.
>
> > Experiments. MaxEnt performance
>
> In the MaxEnt paper, the evaluation was done by computing the state visitation distribution across multiple trajectories. However, in this work, we focused on learning policies that can optimize the entropy and attain optimal coverage within a single trajectory. In Appendix H.2, we evaluated MaxEnt across multiple trajectories and showed that the performance of MaxEnt improves with the number of trajectories. However, on Reacher is a harder task to solve as the state space and action space are complex compared to the grid-based environment, we see that the performance of MaxEnt improves during training (Figure 4(a) in the main paper). Furthermore, a random policy was found to have small coverage of the state space even across multiple trajectories as the agent starts at one of the corners in the state space and needs to be efficient to cover the state space.
>
> > How the approach relates to Forward-Backward representations?
>
> Thank you for pointing out this connection. Intuitively, the Forward-Backward representations capture similar state visitation statistics as the predecessor and successor representation used by $\eta\psi$-Learning. Mathematically, the forward-backward (FB) representation factorizes the Q-function in an RL setting in a very different way than the $\eta\psi$-Learning algorithm. The FB method also focuses on a reward-maximization setting instead of exploration and are only conditioned on states whereas the $\eta\psi$-Learning algorithm conditions these representations on trajectories. During training, the focus of FB learning is on representation learning using an offline dataset where it is assumed that the agent does not have to explore the environment. The learned FB representations are then used to solve multiple different tasks using the same representations, assuming that reward parameterization is known. However, the $\eta\psi$-Learning algorithm learns exploratory policies that could be used to initially explore an unknown task to determine this reward parameterization. As suggested by the comment, the $\eta\psi$-Learning algorithm complements the FB method. We believe that integrating the two systems into a cohesive RL agent is an interesting avenue for future research and have added this discussion to the paper.
>
> > The paper does not include an explicit assessment of the limitations
>
> We have added a section titled limitations in the paper. We mention the points below in brief and have expanded on them in the paper.
> 1. *Scaling to high-dimensional inputs:* We used the same points from the discussion above on scaling the method.
> 2. *Environments with changing dynamics:* Currently, the method is limited to exploring within the same environment where transition dynamics do not change, and extending to procedural environments is hard as it requires SR vectors that can adapt to the changes in the environment and we leave it for future research.
> 3. *Architectural priors for estimating SR:* The successor representations use a GRU to encode prior states and future research can explore having better architectures like Transformers or S4 that have better long-term memory.
> 4. *Estimating predecessor representation:* Future research can also explore learning predecessor representation vectors as done in [4] to improve sample efficiency.
>
> In addition to the limitations, we have added the points on state visitation instead of entropy and connection with FB representations in the discussions. We hope we addressed the concerns and would be happy to take more questions. We hope the reviewer will consider increasing the score.
>
> #### References
> [1] Mutti et al., "Task-agnostic exploration via policy gradient of a non-parametric state entropy estimate." AAAI’21.\
> [2] Touati et al., "Learning one representation to optimize all rewards." NeurIPS’21.\
> [3] Farebrother et al., "Proto-value networks: Scaling representation learning with auxiliary tasks." ICLR’23.\
> [4] van Hasselt et al., "Expected eligibility traces." AAAI’21.

---

> > ### Author Response · Authors · 2023-08-14
> >
> > We thank the reviewer again for their review. We have added ways to scale the proposed method and added more baselines. We also discussed on comparison with FB representation and estimation of state entropy instead of state visitation and updated the paper with these points.
> >
> > As the end of the discussion period is approaching, we would like to kindly ask you to review the changes and assess whether the concerns are addressed. If so, we hope that you would be willing to increase your score.
> >
> > Thank you for your time,\
> > The Authors

---

> > > ### Author Response · Authors · 2023-08-18
> > >
> > > Dear Reviewer RqRN,
> > >
> > > We hope that you've had a chance to read our rebuttal. As the end of the discussion period is approaching, we would greatly appreciate a reply as to whether our response and clarifications have addressed the issues raised in your review.
> > >
> > > Thank you for your time,\
> > > The Authors

---

> > ### Comment · Reviewer_RqRN · 2023-08-19
> > **After response**
> >
> > I am very sorry for my late reply.
> >
> > I want to thank the authors for their detailed comments on the points I raised. I am happy to keep my original positive evaluation, and I will recommend accepting this paper in the private discussion.

---

### Official Review · Reviewer_gBvc · 2023-06-28

**Soundness:** 4 excellent
**Presentation:** 4 excellent
**Contribution:** 3 good
**Rating:** 8
**Confidence:** 4

**Summary:**

This paper shows a combination of "succesor" and "predecessor" representations can be used to develop an efficient maximum entropy exploration policy.

**Strengths:**

- Overall, the paper is clearly written and makes a useful contribution to the exploration literature.

- As far as I know, the ideas are novel.

- The authors make a strong empirical case for their algorithm.

- I'm a bit unsure about the significance of the algorithm, beyond the empirical results shown in the paper. I'm not sure whether the paper is sufficiently ground-breaking to have a significant impact on the broader reinforcement learning literature.

**Weaknesses:**

- I think the authors are a bit loose with their arguments about human cognition. It is hotly debate to what extent cognition depends on language in a strong way. It also important to note that one can endorse a "language of thought" hypothesis about high-level cognition without endorsing the hypothesis that this corresponds to natural language. In any case, I appreciate that these points have little bearing on the substance of this paper.

- Eq. 1 could benefit from more explanation.

- Please state what error bars show in figures.

UPDATE: the authors have addressed my comments.

**Questions:**

The authors should be able to easily address my releatively minor comments in the "weaknesses" section.

More broadly, I think the general usefulness of this approach will depend on how it can be applied beyond the maximum entropy exploration setting.

UPDATE: the authors have addressed my comments.

**Limitations:**

In the Discussion, the authors discuss several ways to improve and extend their algorithm.

---

> ### Author Rebuttal · Authors · 2023-08-09
>
> We thank the reviewer for their time and valuable feedback. We aim to address the concerns in the following:
> > I think the authors are a bit loose with their arguments about human cognition.
>
> We thank the reviewer for bringing this up and agree with the fact that it is not clear how cognition depends on language. As mentioned by the reviewer, we have just mentioned them as part of conceptual motivation and focused on the problem of designing novel methods for efficient exploration.
> > Eq. 1 could benefit from more explanation.
>
> We thank the reviewer for pointing this out. To clarify this, we have adjusted the writing between lines 83-89 to:\
> For a trajectory $\tau=(s_1,a_1,...,a_{h-1},s_h)$ of length $h$, we want to compute the state visitation distribution to estimate the entropy. Each state within the trajectory can be formally expressed as a vector by first encoding the state $s_t$ as a one-hot bit vector $e_{s_t}$. The h-step state visitation distribution for $\tau_h$ can be computed by marginalizing across the time steps:
> $$\xi_{\gamma,\tau} = \sum_{t=1}^h \gamma(t) e_{s_t},(Eq. 1)$$
> where $\gamma: \mathbb{N} \to [0, 1]$ is the *discount function*  (we denote the set of positive integers with $\mathbb{N}$), such that $\sum_{t=1}^h \gamma(t)=1$. Using the normalization in the discount function is necessary as it ensures that $\xi_{\gamma,\tau}$ is a probability vector. We note that this use of a discount function is distinct from using a discount factor in common RL algorithms such as Q-learning but using a discount function is necessary as we will elaborate in the following section. The expected state visitation distribution for a policy $\pi$ can be obtained by generating multiple trajectories using $\pi$ and computing the average across them to get the expected state visitation distribution, denoted by $E_{\tau} [\xi_{\gamma,\tau}]$.
>
> > Please state what error bars show in the figures.
>
> We mentioned in line 237 in the paper that the error bars denote the 95% confidence interval. However, we agree with the reviewer and have added this detail in the caption of the figures for more clarity.
>
> > I think the general usefulness of this approach will depend on how it can be applied beyond the maximum entropy exploration setting.
>
> We agree with the reviewer that the proposed method will have more impact when applied to broader scenarios. Prior methods on maximum state entropy exploration were optimized to have optimal entropy across multiple trajectories of long length. However, to extend such methods to general tasks there is a need for more efficient policies that can explore the state space with minimal interactions. We believe $\eta\psi$-Learning by learning to explore efficiently within a single episode of limited length is a step towards using such policies for standard RL tasks. In the paper, we demonstrate the comparison of $\eta\psi$-Learning with Meta-RL methods where exploration is required at the beginning to find rewarding states efficiently during evaluation (Figure 4(c)).
>
> We also added an additional experiment on a sparse MountainCar environment where the agent is rewarded only after reaching the goal state. Such tasks demand exploration at the start to gather rewarding transitions to improve sample efficiency. We compare with the following baselines: random agent,  TD3 [1], TD3 combined with count-based bonus as an intrinsic reward (TD3-Count), and TD3 combined with first occupancy bonus [2] as an intrinsic reward (TD3-First). We also propose a variant of our method combined with TD3 and call it TD3-$\eta\psi-Learning$, which learns 2 critics- one to estimate the SR as mentioned in Algorithm 2 ($Q_{expl}$), and the other to estimate the Q-function conditioned on the current state as done in TD3 ($Q_{ext}$). Analogous to TD3, the latter critic is learned using extrinsic rewards. Lastly, to update the actor, the agent optimizes for the overall Q-function which is defined as the linear sum of both Q-values based on extrinsic rewards and entropy-based term: $Q_{total} = Q_{ext} + \beta Q_{expl}$.
>
> Figure 4 In the rebuttal pdf presents the results on Sparse MountainCar environment, where the proposed method outperforms the baselines. We have added this experiment with a pseudo-code for the TD3-$\eta\psi$-Learning and leave it for future work to leverage this approach for harder exploration tasks.
>
> We hope we addressed most of the concerns and hope you would be willing to increase your score.
>
> #### References
> [1] Fujimoto et al., "Addressing function approximation error in actor-critic methods." ICML’18.\
> [2] Moskovitzet al., "A First-Occupancy representation for reinforcement learning.", ICLR’22

---

> > ### Comment · Reviewer_gBvc · 2023-08-14
> > **Response to rebuttal**
> >
> > I thank the authors for comprehensively responding to my comments. I feel that the paper should be accepted, and I am raising my score to 8.

---

### Official Review · Reviewer_L5xE · 2023-07-05

**Soundness:** 2 fair
**Presentation:** 3 good
**Contribution:** 2 fair
**Rating:** 5
**Confidence:** 5

**Summary:**

This paper proposes a new exploration method under maximum entropy RL settings. At each time step, the agent selects the action that maximises the expected entropy of the finite-length trajectory. The trajectory entropy is decomposed into two terms, based on variants of the predecessor representation and successor representation, respectively. The authors proposed separate training frameworks under discrete and continuous action spaces. The resulting agent is evaluated in grid worlds with different configurations.

**Strengths:**

- The proposed decomposition of trajectory entropy objective is novel;
- The authors provide comprehensive description of Q-learning and policy gradient training under discrete and continuous action spaces, respectively.
- Empirical evaluations are comprehensive and coheres with the arguments in the paper;

**Weaknesses:**

- The predecessor representation part (for the computation of the entropy of past trajectory) seems unnecessary and does not contribute to action selection, could the authors elaborate on this point?
- The evaluations in continuous control tasks is still within the grid world environment, could the authors evaluate the proposed agent in standard exploration-demanding continuous control tasks;

**Questions:**

See questions in Weaknesses.

**Limitations:**

Yes.

---

> ### Author Rebuttal · Authors · 2023-08-09
>
> We thank the reviewer for their time and valuable feedback. We aim to address the concerns in the following:
> > The predecessor representation part (for the computation of the entropy of past trajectory) seems unnecessary and does not contribute to action selection, could the authors elaborate on this point?
>
> The entropy is dependent on the state visitation distribution which is computed across the entire trajectory in the episode. At any time (T), the agent has access to the history of states and needs to take action to maximize the entropy of the state visitation distribution. Since the visited states till time T are known to the agent, they are used to compute the visitation distribution for the first T time steps resulting in the predecessor representation vector (Equation 5). To estimate the distribution of the future states, the agent predicts the expected visitation distribution of future states—the successor representation vector as defined in Equation 6. The predecessor representation is used to encode the visitation distribution of past states and the successor representation is used to predict the visitation distribution of future states (see our example in Figure 1). In Equation 7, we propose to aggregate them, which gives the expected state visitation distribution for the whole trajectory. The Q-function is defined as the entropy of this estimated state visitation distribution. Although the successor representation is conditioned on the previously visited states, we need predecessor representation to calculate this expected state visitation distribution and thus the state visitation entropy of a single trajectory, as stated in Equation 7. Since the agent selects an action based on the Q-function, the action selection is conditioned on the state visitation distribution which further depends on the predecessor representation. We have expanded on this in the revised version of the paper to explain the importance of predecessor representation in the action selection.
>
> > The evaluations in continuous control tasks is still within the grid world environment, could the authors evaluate the proposed agent in standard exploration-demanding continuous control tasks;
>
> The experiments on continuous control tasks were conducted on the state space provided by the environment. The input to the recurrent architecture~(Figure 5 in Appendix) is the continuous 8-dimensional and 27-dimensional vector obtained from Reacher and Pusher environments, respectively. The predecessor and successor representation is computed over the discretized vector obtained using the (x, y) coordinates of the fingertip. To further show that the method can scale to harder continuous control tasks where we would want to compute the entropy over a larger state space, we conducted an experiment on HalfCheetah environment. Taking inspiration from ProtoValueNetworks [1] to avoid the curse of dimensionality while discretization, we discretize each dimension of the state space separately and compute the overall entropy by averaging the entropy across all dimensions. The successor and predecessor representations are computed on the 17-dimensional state space, where each part is discretized into 10 bins for our experiments. Also, the state coverage in evaluation metrics is computed as the average coverage across the discretized dimension. We report the results in Figure 2 in rebuttal PDF where the results are obtained across 5 seeds. We compare with MEPOL [2] which uses Markovian policy and uses kNN to estimate the entropy. $\eta\psi$-Learning was found to have better coverage and entropy over the discretized state space demonstrating that the proposed discretization can be leveraged to scale the proposed method to large state space. We leave it for future research to extend the method to explore POMDPs with more complex environments, including images and have added these discussions to the revised version of the paper.
>
> We thank the reviewer for the feedback. We hope we addressed most of your questions, and hope you consider updating your score.
>
> #### References
> [1] Farebrother et al., "Proto-value networks: Scaling representation learning with auxiliary tasks." ICLR’23\
> [2] Mutti et al., "Task-agnostic exploration via policy gradient of a non-parametric state entropy estimate." AAAI’21.

---

> > ### Author Response · Authors · 2023-08-14
> >
> > We thank the reviewer again for their review. We have elaborated on the importance of predecessor representation and present how the proposed method can be scaled to tasks with more complex state spaces. We would be happy to answer any further questions.
> >
> > As the end of the discussion period is approaching, we would like to kindly ask you to review the changes and assess whether the concerns are addressed. If so, we hope that you would be willing to increase your score.
> >
> > Thank you for your time,\
> > The Authors

---

> > > ### Author Response · Authors · 2023-08-18
> > >
> > > Dear Reviewer L5xE,
> > >
> > > We hope that you've had a chance to read our rebuttal. As the end of the discussion period is approaching, we would greatly appreciate a reply as to whether our response and clarifications have addressed the issues raised in your review.
> > >
> > > Thank you for your time,\
> > > The Authors

---

### Official Review · Reviewer_RPnY · 2023-07-07

**Soundness:** 2 fair
**Presentation:** 3 good
**Contribution:** 2 fair
**Rating:** 6
**Confidence:** 4

**Summary:**

This paper proposes a novel exploration algorithm in RL by combining the successor representation with the predecessor representation and maximising episode-level entropy of state visitation. The proposed approach demonstrates improvement over the MaxEnt baseline on simple Gridworld and continuous control environments.

**Strengths:**

- This paper presents a nice way to bridge the successor and predecessor representation to maximise entropy in the visitation of states in an episode.
- The paper is clearly written and well motivated.

**Weaknesses:**

- In the experiments, the proposed approach is only compared with the MaxEnt baseline. While this is a good baseline to compare with since the objective is similar, the standard exploration baselines are missing: 1) epsilon-greedy exploration, 2) count-based intrinsic motivation approaches like UCB, 3) random action baseline, and optionally 4) auxiliary objectives for exploration, like curiosity-based learning. While I agree with the related work section that these works learn Markovian policies while the proposed work learns a policy conditioned on the full history, a couple of them should still have been included, to place this algorithm in the overall landscape of exploration algorithms.
- Since the successor representation and the predecessor representation in this case both depend on the full history, it is not possible to disentangle the impact they have on exploration and the evaluation metrics as there is no baseline included in which the successor representation depends only on the current state.
- A relevant paper that is missing from the related work section, and also from the baselines, is "A First-Occupancy Representation for Reinforcement Learning" by Moskovitz et al, which is another state representation that indicates the time of first access of a state by the agent and has been shown to be conducive to exploration.


**Questions:**

Please refer to the comments listed in the Weaknesses section.

**Limitations:**

The authors have discussed general high-level ethical concerns with this line of work, but the limitations of the proposed approach have not been discussed.

---

> ### Author Rebuttal · Authors · 2023-08-09
>
> We thank the reviewer for their time and valuable feedback. We aim to address the concerns in the following:
> > In the experiments, the proposed approach is only compared with the MaxEnt baseline.
>
> We thank the reviewer for pointing this out. We did an experiment with the sparse MountainCar environment (Figure 4 in Rebuttal PDF) and showed that the proposed method promotes exploration in sparse reward environments. We have described this experiment in more detail in the General Response section. Furthermore, we did not compare intrinsic curiosity and count-based methods on reward-free tasks as such methods do not learn exploratory policies at convergence. This is because the intrinsic curiosity or the count-based bonus becomes zero or uniform at convergence. This was also discussed in Section 2 in SMM [1], where they elaborate on the difference between such methods and the current method’s line of work. We agree with the reviewer that the exploration algorithm based on maximum state entropy exploration should be compared with the prior works on exploration and hope the MountainCar experiment presents the benefit of the proposed method. Lastly, we have also added MEPOL [2] which learns a Markovian policy and uses a non-parametric entropy estimator to learn optimal policies for continuous control tasks, and a random policy as additional baselines (Figure 3 in rebuttal PDF).
>
> > There is no baseline included in which the successor representation depends only on the current state.
>
> As suggested by the reviewer, we conduct experiments on Reacher and Pusher environments where the successor representation is conditioned over the current state only. However, the predecessor representation is only used to compute the state visitation distribution which is necessary for the objective function. In Figure 3 of rebuttal PDF, we can observe that the modification does not perform well on both these tasks, and learning SR conditioned on the history of visited states is necessary for learning optimal behaviors.
>
> > A relevant paper that is missing from the related work section, and also from the baselines, is "A First-Occupancy Representation for Reinforcement Learning"
>
> We thank the reviewer for pointing this out. We were unaware of this work and have added this paper to the related works section. Furthermore, in the experiment on Sparse MountainCar, we also used the First Occupancy-based intrinsic reward as the intrinsic bonus in each episode for comparison, where the proposed method was found to perform better.
>
> > The authors have discussed general high-level ethical concerns with this line of work, but the limitations of the proposed approach have not been discussed.
>
> In this work, we focus on developing a method that can be optimized with maximum state entropy objective and learn to explore within a single episode of finite length. As with any new approach, there are certain limitations:
> 1. *Scaling to high-dimensional inputs:* Learning to explore more complex tasks with high-dimensional input spaces would require using a better representation learning method and a mechanism to estimate successor and predecessor representations. For representation learning, existing methods that use auxiliary losses, inverse/forward dynamics, or random network-based features can be used. The more challenging task is learning the SR and future works can explore leveraging methods like Successor Measures[4] or ProtoValueNetworks[4].
> 2. *Environments with changing dynamics:* The learned SR depends on the environment dynamics and the policy, and we learn SR for a fixed environment in this work.  However, many real-world tasks require exploration in an environment with changing dynamics~(procedural environments [5]). A potential direction is learning universal successor representation approximators [6] where the successor representations are conditioned on a context that defines the environment and we leave this for future research.
> 3. *Architectural priors for estimating SRs:*  The successor representations use an RNN which is known to suffer from vanishing gradient problems. Many real-world tasks require agents to retain information over multiple timesteps. Future research can explore having better architectural priors like Transformers or S4 that have better memory and are known to work well on complex tasks.
> 4. *Estimating predecessor representation:* In this work, we computed the predecessor representation as the summation of the prior state representations. However, recent methods like Expected Eligibility Traces [7] show better sample efficiency and we leave leveraging such methods for future research.
>
>
> Lastly, we have added these limitations to a separate section in the paper. We hope we addressed most of your questions, and hope you consider updating your score.
>
> #### References
> [1] Lee, et al., "Efficient exploration via state marginal matching." arXiv preprint arXiv:1906.05274 (2019).\
> [2] Mutti et al., "Task-agnostic exploration via policy gradient of a non-parametric state entropy estimate." AAAI’21.\
> [3] Touati et al., "Learning one representation to optimize all rewards." NeurIPS’21.\
> [4] Farebrother et al., "Proto-value networks: Scaling representation learning with auxiliary tasks." ICLR’23.\
> [5] Zha et al., "Rank the episodes: A simple approach for exploration in procedurally-generated environments." ICLR’23.\
> [6] Borsa et al. "Universal successor features approximators." arXiv preprint arXiv:1812.07626 (2018).\
> [7] van Hasselt et al., "Expected eligibility traces." AAAI’21.

---

> > ### Author Response · Authors · 2023-08-14
> >
> > We thank the reviewer again for their review. We have added more baselines, an experiment where the SR is only conditioned on the current state, and also experimented in a sparse reward environment to compare with other exploration-based methods.
> >
> > As the end of the discussion period is approaching, we kindly ask the reviewer to engage in further discussion and we hope the reviewer could adjust their score accordingly if all raised concerns are addressed.
> >
> > Thank you for your time,\
> > The Authors

---

> > > ### Comment · Reviewer_RPnY · 2023-08-17
> > >
> > > I thank the authors for addressing the points I raised in my review, and have increased my score.

---

### Author Rebuttal · Authors · 2023-08-09

Firstly, we thank the reviewers for their time and constructive feedback. We hope to address the concerns during the rebuttal and would be happy to answer more questions.

In this work, we developed an algorithm to learn exploratory policies at convergence that can explore the state space efficiently within a finite-length trajectory. Such policies can benefit generalization in different applications like Meta-RL and episodic exploration. Maximum state entropy exploration is a potential direction for learning such policies. However, prior works are not very efficient as they either learn a Markovian policy, optimize for the state coverage over multiple long trajectories, or learn a mixture of stochastic policies. Due to these shortcomings, they are not widely used for solving tasks in RL. To address these concerns, we introduce $\eta\psi$-Learning and demonstrate that the proposed algorithm can learn to efficiently explore the state space within a finite length trajectory. $\eta\psi$-Learning achieves this by combining predecessor and successor representation to estimate the state-visitation distribution and utilizing this to optimize the entropy-based objective. Mutti et al.’22 [1] theoretically showed that learning such policies that achieve zero-regret is NP-Hard and we develop a practical algorithm to solve such tasks. We hope that the proposed method bridges the gap of leveraging policies learned using maximum state entropy exploration for more complex tasks in RL.

In Figure 4(c), we show that when compared with VariBAD, $\eta\psi$-Learning is more efficient at finding the reward function. We are adding a few more experiments to demonstrate the broader applicability of $\eta\psi$-Learning (discussed below):
1. To compare with other exploration algorithms in standard RL tasks, we performed additional experiments on the Sparse MountainCar environment. The agent receives a positive reward after reaching the goal position after which the episode terminates and no reward in other states. We chose this environment because the reward is sparse, and it is hard for the agent to discover the reward function as the agent needs to plan to explore the top of the hill. For the baselines, we compare with a random agent,  TD3 [2], TD3 combined with count-based bonus as an intrinsic reward (TD3-Count), and TD3 combined with first occupancy bonus [5] as an intrinsic reward (TD3-First).\
We also propose a variant of our method combined with TD3 and call it TD3-$\eta\psi$-Learning. For this, we propose to learn 2 critics- one to estimate the SR as mentioned in Algorithm 2 ($Q_{expl}$), and the other to estimate the sum of extrinsic rewards conditioned on the current state similar to TD3 ($Q_{ext}$). Lastly, to update the actor, the gradients are obtained using the overall Q-function which is defined as the linear sum of both Q-values based on extrinsic rewards and entropy-based term: $Q_{total} = Q_{ext} + \beta Q_{expl}$. We have added a pseudo-code of this algorithm in the Appendix of the paper.\
Figure 4 in the rebuttal pdf presents the results on the Sparse MountainCar environment. We compare two metrics- Return and Average Steps taken to reach the goal state across 5 seeds. The Average Steps metric highlights if the agent learns to solve the task with minimal interactions. We plot the mean and 95% confidence interval in shading. Through our experiment on Sparse MountainCar, we demonstrate that the proposed method can improve efficiency in standard RL tasks, especially in sparse reward environments. Future works can leverage the proposed extension in POMDP setting with high-dimensional inputs like images.
2. In this experiment, we wanted to show that $\eta\psi$-Learning can be scaled to high-dimensional space and conduct an experiment on the HalfCheetah environment. We propose to learn successor representations for larger state spaces by using ideas similar to ProtoValueNetworks [3]. To learn SR and predecessor representation, we discretize each dimension of the state space into K bins. Thus, a continuous state can be converted to |S| one-hot vectors where each vector is of K dimensions. The overall entropy and coverage are calculated by averaging the entropy and coverage over each dimension of the state space. We have used MEPOL [4] as the baseline and used the author’s implementation for our experiments.  We observed $\eta\psi$-Learning outperforms MEPOL on entropy and coverage metrics when evaluated over a single trajectory of 1000 steps~(Figure 2 of rebuttal pdf). Through this experiment, we wanted to show that the proposed method can be scaled to larger state spaces and believe future work can explore scaling them to more complex environments.

In the above experiments, we show that the method can be applied to tasks with rewards and can be scaled to environments with high-dimensional state spaces and leave it for future work to scale to more complex environments.

We hope we addressed most of the concerns during the rebuttal and would be happy to answer further questions.

#### References
[1] Mutti et al., "The importance of non-markovianity in maximum state entropy exploration." ICML’22.\
[2] Fujimoto et al., "Addressing function approximation error in actor-critic methods." ICML’18.\
[3] Farebrother et al., "Proto-value networks: Scaling representation learning with auxiliary tasks." ICLR’23\
[4] Mutti et al., "Task-agnostic exploration via policy gradient of a non-parametric state entropy estimate." AAAI’21.\
[5] Moskovitz et al., "A First-Occupancy representation for reinforcement learning." ICLR’22.

---

> ### Author Response · Authors · 2023-08-12
>
> We thank the reviewers for their time and feedback on improving our work. We are delighted that the reviewers find the decomposition of estimating state visitation distribution using successor and predecessor representations and proposed learning frameworks in Q-learning and policy gradient setting novel.
>
> We have attempted to address all concerns and significantly improved the manuscript as a result. Based on the feedback, we have 1) added baselines on continuous control tasks, 2) presented an experiment to show that the method can be scaled, and 3) shown an application where $\eta\psi$-Learning combined with existing methods can learn efficiently in sparse reward environment.
>
> We would be happy to answer any questions during the discussion period and hope the reviewers would be willing to increase the score.
>
> Thank you for your time,
>
> The Authors

---

### Decision · Program_Chairs · 2023-09-21

**Decision:**

Accept (poster)

**Comment:**

This paper proposes an maximum state entropy exploration method, called $\eta \psi$-learning. The idea is to maximize the state entropy of finite length trajectories and to combine predecessor and successor representations.

Reviewers appreciate the novelty and clear presentation of the idea, as well as the good empirical performance on toy domains and some continuous control tasks comparing with MaxEnt. Concerns regarding more exploration methods should be taken as baselines were answered by authors providing additional experiments on Sparse MountainCar environment. Concerns of how the proposed method could be scaled and perform in higher dimensional settings were partially addressed by additional results on HalfCheetah.

While reviewers agreed that the paper makes a good contribution to exploration methods, we recommend the authors to include also the discussions and clarifications about the importance of predecessor, limitations, as well as how to scale up the methods to more complex scenarios into next versions.